# Preexisting memory CD4 T cells in naïve individuals confer robust immunity upon hepatitis B vaccination

George Elias[1,2]*[†], Pieter Meysman[2,3,4][†], Esther Bartholomeus[2,5][†], Nicolas De Neuter[2,3,4], Nina Keersmaekers[2,6], Arvid Suls[2,5], Hilde Jansens[7], Aisha Souquette[8], Hans De Reu[1,9], Marie-Paule Emonds[10], Evelien Smits[1,9], Eva Lion[1,9], Paul G Thomas[8], Geert Mortier[2,5], Pierre Van Damme[2,11], Philippe Beutels[2,6], Kris Laukens[2,3,4][†], Viggo Van Tendeloo[1†‡], Benson Ogunjimi[2,6,12,13]*[†]

[1]Laboratory of Experimental Hematology (LEH), Vaccine and Infectious Disease Institute, University of Antwerp, Antwerp, Belgium; [2]Antwerp Unit for Data Analysis and Computation in Immunology and Sequencing, University of Antwerp, Antwerp, Belgium; [3]Adrem Data Lab, Department of Mathematics and Computer Science, University of Antwerp, Antwerp, Belgium; [4]Biomedical Informatics Research Network Antwerp, University of Antwerp, Antwerp, Belgium; [5]Department of Medical Genetics, University of Antwerp, Antwerp, Belgium; [6]Centre for Health Economics Research & Modeling Infectious Diseases, Vaccine & Infectious Disease Institute (VAXINFECTIO), University of Antwerp, Antwerp, Belgium; [7]Department of Clinical Microbiology, Antwerp University Hospital, Antwerp, Belgium; [8]Department of Immunology, St. Jude Children's Research Hospital, Memphis, United States; [9]Center for Cell Therapy and Regenerative Medicine, Antwerp University Hospital, Antwerp, Belgium; [10]Histocompatibility and Immunogenetic Laboratory, Rode Kruis-Vlaanderen, Mechelen, Belgium; [11]Centre for the Evaluation of Vaccination (CEV), Vaccine and Infectious Disease Institute, University of Antwerp, Antwerp, Belgium; [12]Antwerp Center for Translational Immunology and Virology (ACTIV), Vaccine and Infectious Disease Institute, University of Antwerp, Antwerp, Belgium; [13]Department of Paediatrics, Antwerp University Hospital, Antwerp, Belgium

*For correspondence:
igeorgeelias@gmail.com (GE);
benson.ogunjimi@uantwerpen.be (BO)

[†]These authors contributed equally to this work

Present address: [‡]Janssen Research and Development, Immunosciences WWDA, Johnson and Johnson, Beerse, Belgium

**Abstract** Antigen recognition through the T cell receptor (TCR) αβ heterodimer is one of the primary determinants of the adaptive immune response. Vaccines activate naïve T cells with high specificity to expand and differentiate into memory T cells. However, antigen-specific memory CD4 T cells exist in unexposed antigen-naïve hosts. In this study, we use high-throughput sequencing of memory CD4 TCRβ repertoire and machine learning to show that individuals with preexisting vaccine-reactive memory CD4 T cell clonotypes elicited earlier and higher antibody titers and mounted a more robust CD4 T cell response to hepatitis B vaccine. In addition, integration of TCRβ sequence patterns into a hepatitis B epitope-specific annotation model can predict which individuals will have an early and more vigorous vaccine-elicited immunity. Thus, the presence of preexisting memory T cell clonotypes has a significant impact on immunity and can be used to predict immune responses to vaccination.

## Editor's evaluation

By using modern high-throughput sequencing this paper demonstrates that the antibody mediated immune responses that are elicited by vaccination are improved by pre-existing memory CD4 T cell responses. The experimental data are an important contribution and would be useful as a data resource for future research. All reviewers agree that the findings are of great interest to the community.

## Introduction

Antigen recognition through the T cell receptor (TCR) is one of the key determinants of the adaptive immune response (*Rudolph et al., 2006*). Antigen presentation via major histocompatibility complex (MHC) proteins, together with the right costimulatory and cytokine signals, are responsible for T cell activation (*Curtsinger and Mescher, 2010*; *Esensten et al., 2016*). The TCR αβ heterodimer binds the peptide-MHC (pMHC) complex and confers the specificity of a T cell to an epitope. A highly diverse TCR repertoire ensures that an effective T cell response can be mounted against pathogen-derived peptides (*Turner et al., 2009*). High TCRαβ diversity is generated through V(D)J recombination at the complementary-determining region 3 (CDR3) of TCRα and TCRβ chains, accompanied with junctional deletions and insertions of nucleotides, further adding to the diversity (*Krangel, 2009*).

Vaccines activate naïve T cells with high specificity to vaccine-derived peptides and induce T cell expansion and differentiation into effective and multifunctional T cells. This is followed by a contraction phase from which surviving cells constitute a long-lived memory T cell pool that allows for a quick and robust T cell response upon a second exposure to the pathogen (*Farber et al., 2014*). However, previous studies have shown that a prior pathogen encounter is not a prerequisite for the formation of memory T cells and that CD4 T cells with a memory phenotype can be found in antigen-naïve individuals (*Su et al., 2013*). The existence of memory-like CD4 T cells in naïve individuals (*Sewell, 2012*) can be explained by molecular mimicry, as the encounter with environmentally derived peptides activates cross-reactive T cells due to the flexible nature of CD4 T cell recognition of the pMHC complex (*Wilson et al., 2004*). Indeed, work that attempted to replicate the history of human pathogen exposure in mice has shown that sequential infections altered the immunological profile and remodeled the immune response to vaccination (*Reese et al., 2016*). The existence of memory CD4 T cells specific to vaccine-derived peptides in unexposed individuals might confer an advantage in vaccine-induced immunity. In the present study, we used high-throughput sequencing to profile the memory CD4 TCRβ repertoire of healthy hepatitis B (HB)-naïve adults before and after administration of an HB vaccine to investigate the impact of preexisting memory CD4 T cells on the immune response to the vaccine. Based on anti-hepatitis B surface (anti-HBs) antibody titers over 365 days, vaccinees were grouped into early-, late-, and non-converters. Our data reveals that individuals with preexisting vaccine-specific CD4 T cell clonotypes in the memory CD4 compartment had earlier emergence of antibodies and mounted a more vigorous CD4 T cell response to the vaccine. Moreover, we identify a set of vaccine-specific TCRβ sequence patterns which can be used to predict TCR specificity and in turn, which individuals will have an early and more vigorous response to HB vaccine.

## Results

### Vaccinee cohort can be classified into three groups

Out of 34 vaccinees, 21 vaccinees seroconverted (an anti-HBs titer above 10 IU/ml was considered protective; *Keating and Noble, 2003*) at day 60 and were classified as early-converters; nine vaccinees seroconverted at day 180 or day 365 and were classified as late-converters; remaining four vaccinees had an anti-HBs antibody titer lower than 10 IU/ml at all time points following vaccination and were classified as non-converters (*Figure 1* and *Figure 1—figure supplement 1A*).

Members of Herpesviridae family might alter immune responses to vaccines (*Furman et al., 2015*). We found no significant differences in cytomegalovirus (CMV), Epstein-Barr virus (EBV), or herpes simplex virus (HSV) seropositivity between the three groups in our cohort (*Figure 1—figure supplement 1B*). Early-converters were slightly younger than late-converters, and non-converters were notably younger than both early- and late-converters (*Figure 1—figure supplement 1C*).

**eLife digest** Immune cells called CD4 T cells help the body build immunity to infections caused by bacteria and viruses, or after vaccination. Receptor proteins on the outside of the cells recognize pathogens, foreign molecules called antigens, or vaccine antigens. Vaccine antigens are usually inactivated bacteria or viruses, or fragments of these pathogens. After recognizing an antigen, CD4 T cells develop into memory CD4 T cells ready to defend against future infections with the pathogen.

People who have never been exposed to a pathogen, or have never been vaccinated against it, may nevertheless have preexisting memory cells ready to defend against it. This happens because CD4 T cells can recognize multiple targets, which enables the immune system to be ready to defend against both new and familiar pathogens.

Elias, Meysman, Bartholomeus et al. wanted to find out whether having preexisting memory CD4 T cells confers an advantage for vaccine-induced immunity. Thirty-four people who were never exposed to hepatitis B or vaccinated against it participated in the study. These individuals provided blood samples before vaccination, with 2 doses of the hepatitis B vaccine, and at 3 time points afterward. Using next generation immune sequencing and machine learning techniques, Elias et al. analyzed the individuals' memory CD4 T cells before and after vaccination.

The experiments showed that preexisting memory CD4 T cells may determine vaccination outcomes, and people with more preexisting memory cells develop quicker and stronger immunity after vaccination against hepatitis B. This information may help scientists to better understand how people develop immunity to pathogens. It may guide them develop better vaccines or predict who will develop immunity after vaccination.

## Unperturbed diversity in memory CD4 T cell repertoire following vaccination

A genomic DNA-based TCRβ sequence dataset of memory CD4 T cells isolated from peripheral blood was generated from a cohort of 33 healthy vaccinees (see Materials and methods for details) right before vaccination (day 0) and 60 days after administration of the first dose of HB vaccine (30 days after administration of the second vaccine dose).

Between $4.54 \times 10^4$ and $3.92 \times 10^5$ productive TCRβ sequence reads were obtained for each vaccinee at each time point (*Figure 2—figure supplement 1A*). Between 30,000 and 90,000 unique TCRβ sequences were sequenced for each vaccinee at each time point (*Figure 2—figure supplement 1B*). As expected, considering the extremely diverse memory CD4 T cell repertoire (*Klarenbeek et al., 2010*), less than 20% of the TCRβ sequences is shared between the time points for each vaccinee (*Figure 2—figure supplement 1C*).

The diversity of the memory CD4 T cell repertoire of each vaccinee at the two time points was explored, but neither the breadth of the memory CD4 TCRβ repertoire nor the Shannon equitability index was found to have been impacted by the vaccine at day 60 (*Figure 2A*).

## Unique vaccine-specific TCRβ sequences are trackable within memory CD4 T cell repertoire and increase following vaccination

Peripheral blood mononuclear cells from day 60 were labeled with carboxyfluorescein succinimidyl ester (CFSE), a dye that enables tracking of cell proliferation, and stimulated with a pool of peptides spanning HB surface antigen (HBsAg). After day 7 of in vitro expansion, we sorted CFSE^low CD4 T cells (*Becattini et al., 2015*) and extracted mRNA for quantitative assessment of HBsAg-specific TCRβ clonotypes by sequencing (see Materials and methods for details). The establishment of a set of vaccine-specific TCRs enables their tracking within memory CD4 T cell repertoires from day 0 to day 60, based on their CDR3β amino acid sequences.

We detected a significant increase in the breadth of HBsAg-specific TCRβ sequences at day 60 post-vaccination compared to pre-vaccination (mean increase = 107.9%, 95% CI = 62.2–358.5%) (*Figure 2B* and *Figure 2—figure supplement 1D*). This increase far exceeded a baseline control of varicella-zoster virus (VZV)-specific TCRβ sequences established in an identical manner (mean difference = 2.1%, 95% CI = −6.6% to 12.4%). The HBsAg-specific increase was found to be significant when compared

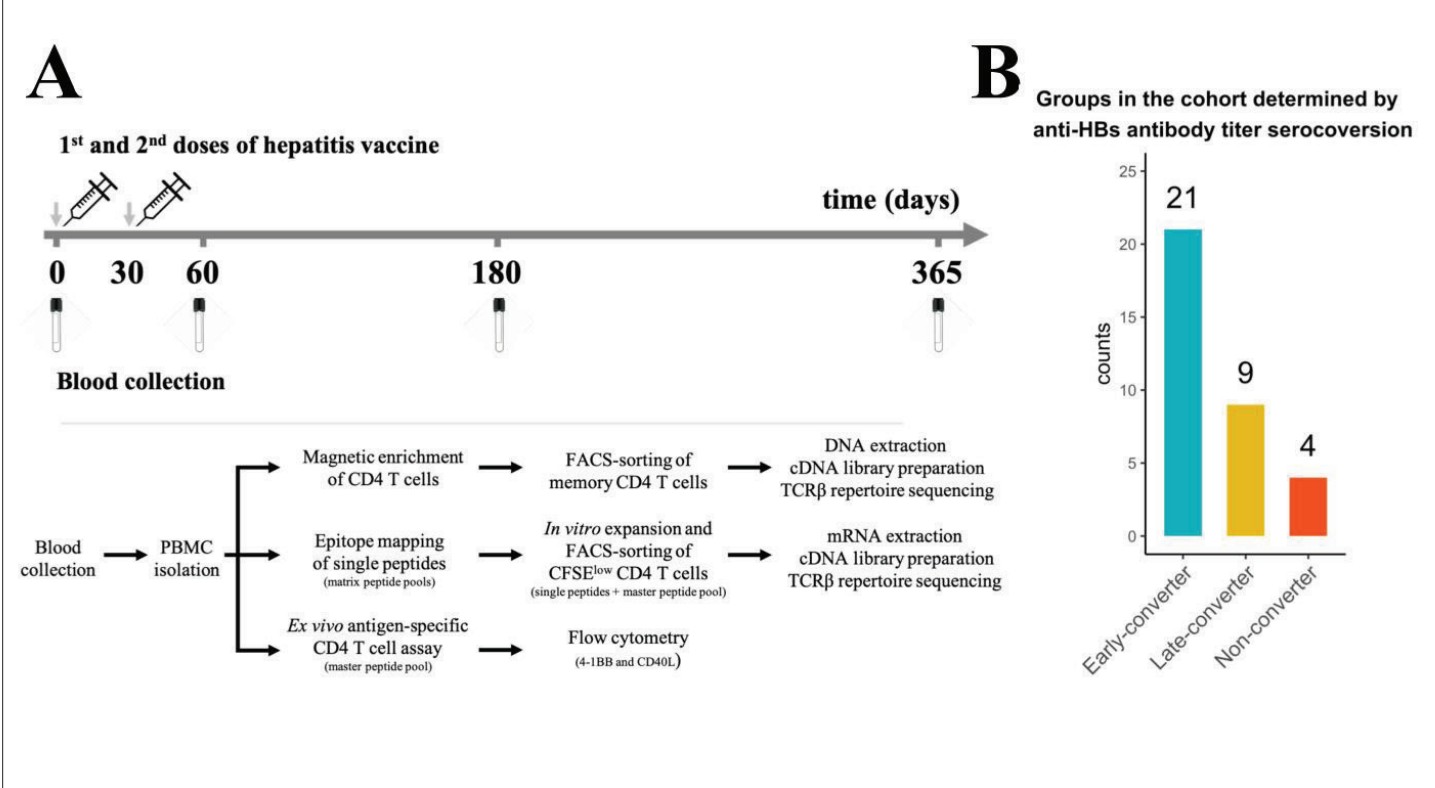

**Figure 1.** Hepatitis B vaccination (Engerix-B) study design. (**A**) Hepatitis B (Engerix-B) vaccination and experimental design. (Top) Timeline of vaccination and blood collection. (Bottom) Memory CD4 T cells were magnetically enriched and FACS-sorted from two time points (day 0 and day 60) for TCRβ repertoire sequencing. Peptide matrix pools were used to map CD4 T cell epitopes of the vaccine from peripheral blood mononuclear cells (PBMCs) collected at day 60 and to select single peptides. After 7 days of in vitro expansion, single peptide-specific and master peptide pool-specific CFSE^low CD4 T cells from PBMCs collected at day 60 were FACS-sorted in two technical replicates for TCRβ repertoire sequencing. PBMCs collected at days 0, 60, 180, and 365 were stimulated with the master peptide pool (HBsAg) and assessed for converse expression of 4-1BB and CD40L by flow cytometry. (**B**) Vaccinee cohort can be classified into three groups as determined by anti-hepatitis B surface (anti-HBs) titer over four times points. Early-converters seroconverted at day 60, late-converters seroconverted at day 180 or day 365, and non–converters did not have an anti-HBs titer higher than 10 IU/ml at any of the time points.

The online version of this article includes the following figure supplement(s) for figure 1:

**Figure supplement 1.** Serological memory to hepatitis B vaccine and vaccinee groups within the cohort.

to this VZV baseline (Wilcoxon p-value 6.3e-05). When considered across the different groups, the increase was found to be significantly different from the VZV baseline for the early-converters (mean = 151.9%, 95% CI = 78.9–342.0%, Wilcoxon Bonferroni-corrected p-value 8.64e-5) and non-converters (mean = 67.6%, 95% CI: 37.6–89.7%, Wilcoxon Bonferroni-corrected p-value 0.03), but not for the late-converters (mean = 18.8%, 95% CI = 3.6–41.8%, Wilcoxon Bonferroni-corrected p-value 0.257).

As HBsAg-specific TCRβ sequences were already detected in the memory CD4 T cell repertoire prior to vaccination, we sought to determine whether the vaccination induced an expansion of those sequences. Based on the frequency of vaccine-specific TCRβ sequences within the memory CD4 T cell repertoire, the data does not support a vaccine-induced expansion of preexisting vaccine-specific TCRβ sequences (*Figure 2—figure supplement 1E*). Thus, although we see a rise in vaccine-specific TCRβ T cells from day 0 to day 60, this cannot be attributed to an expansion of the vaccine-specific TCRβ clonotypes found at day 0 but rather the recruitment of new TCRβ clonotypes, as visualized for one vaccinee in *Figure 2C*.

It makes sense to not only look at the difference in vaccine-specific TCRβ sequences between time points, but also explore whether there are differences in the proportion of HBsAg-specific clones in the memory repertoire between early-converters, late-converters, and non-converters after vaccination. In this case, to allow for a between-vaccinees comparison (in contrast to the within-vaccinees time point comparison), we calculate the overlap coefficient, where HBsAg-specific sequences in

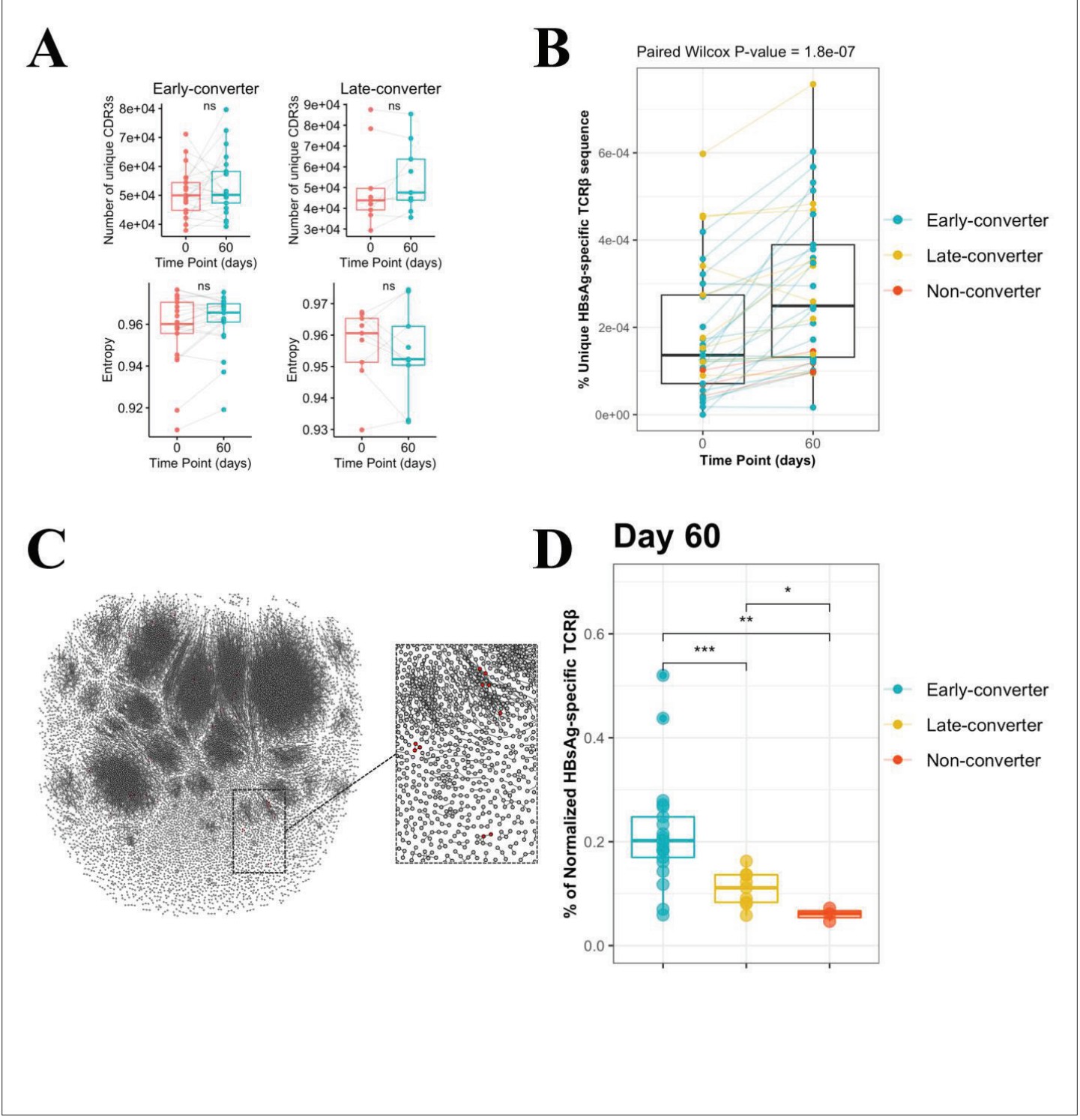

**Figure 2.** CD4 T cell memory T cell receptor β (TCRβ) repertoire and vaccine-specific TCRβ clonotypes. (**A**) Comparison of the memory CD4 TCRβ repertoire diversity, as shown by breadth (number of unique TCRs) and entropy (Shannon equitability index) between day 0 and day 60. Indices are available in *Figure 2—source data 1*. (**B**) Frequency of unique vaccine-specific TCRβ sequences out of total sequenced TCRβ sequences between two time points for all vaccinees colored by group. Frequencies are available in *Figure 2—source data 2*. (**C**) Sequenced CD4⁺ TCR memory repertoire of vaccinee H35 at day 60. Each TCR clonotype is represented by a node. TCRs are connected by an edge if their Hamming distance is one. Only clusters with at least three TCRs are shown. TCR clonotypes in red are the vaccine-specific TCRβ sequences that were not present prior to vaccination. (**D**) Frequency of vaccine-specific TCRβ sequences within memory CD4 T cell repertoire normalized by number of HBsAg-specific TCRβ sequences found for each vaccinee at time point 60. Frequencies are available in *Figure 2—source data 3*.

*Figure 2 continued on next page*

*Figure 2 continued*

The online version of this article includes the following source data and figure supplement(s) for figure 2:

**Source data 1.** Breadth and entropy of T cell receptor β (TCRβ) repertoire.

**Source data 2.** Frequency of unique hepatitis B surface antigen (HBsAg)-specific T cell receptor β (TCRβ) sequences.

**Source data 3.** Frequency of normalized hepatitis B surface antigen (HBsAg)-specific T cell receptor β (TCRβ) sequences.

**Figure supplement 1.** CD4 T cell memory T cell receptor β (TCRβ) repertoire and vaccine-specific TCRβ clonotypes.

the CD4 T cell memory repertoire are normalized by the number of HBsAg-specific TCRβ found for each vaccinee. From this analysis, it can be concluded that there is a difference in HBsAg-specific TCRβ at day 60 between the three groups (*Figure 2D*) (ANOVA p-value = 0.00238). A Wilcoxon test between early-converters and other vaccinees shows a significant p-value of 0.000473, indicating that early-converters have a higher relative frequency of vaccine-specific TCRβ sequences present in their memory CD4 T cell repertoire at day 60 compared to vaccinees from the two other groups in the cohort.

In summary, vaccine-specific TCRβ sequences increased in breadth following vaccination in early-converters and constituted a significantly higher proportion of the memory CD4 T cell compartment post-vaccination.

## HBsAg single peptide-specific TCRβ identification allows predictive modelling of early-converters prior to vaccination

To quantify the T cell response at the level of individual peptides that make up the HBsAg, a peptide matrix designed to cover 54 overlapping peptides of the HBsAg was used to extract peptide-specific T cells using a CD40L/CD154 activation-induced marker (AIM) assay (see Materials and methods for details). The top six peptides for each individual were selected for an in vitro peptide-specific CD4 T cell expansion and sorting for TCR sequencing (*Supplementary file 2* and *Figure 3—figure supplement 1*). In this manner, TCRβ sequences were identified for T cells reactive against 44 single HBsAg peptides. These were not uniformly distributed across the HBsAg amino acid sequence, with the most prominent epitopes covering the regions 1–15, 129–144, 149–164, 161–176, 181–200, 213–228. For each of those regions, more than 10 individuals had a strong T cell response and more than 150 unique TCRβ sequences could be identified (*Figure 3A*).

These peptide-specific TCRβ sequences can be utilized in a peptide-TCR interaction classifier to identify other TCRβ that are likely to react against same HBsAg epitopes, as it has been shown that similar TCRβ sequences tend to target the same epitopes (*De Neuter et al., 2018*; *Meysman et al., 2019*). These classifications were integrated into a model which outputs a ratio $R_{hbs}$, which represents the amount of HBsAg epitope-specific TCR sequences in an individual repertoire. The ratio is equal to the breadth of unique predicted HBsAg peptide-specific TCRβ divided by a normalization term for putative false positive predictions due to bystander activations in the training data set. The memory repertoire at day 60 shows that early-converters tend to have a higher breadth of putative HBsAg peptide-specific TCRβ, while late-converters tend to have relatively more putative false positives as per the normalization term (*Figure 3B*). The ratio of these two terms $R_{hbs}$ therefore shows a significant difference between early- and the late-converters at day 60 (one-sided Wilcoxon-test p-value = 0.0313, *Figure 3C*). Furthermore, searching for HBsAg peptide-specific clonotypes in the memory repertoires prior to vaccination (day 0) results in a $R_{hbs}$ with a similar difference (one-sided Wilcoxon-test p-value = 0.0010, *Figure 3D*). In this manner, the presence of HBsAg peptide-specific clonotypes as defined by the ratio $R_{hbs}$ can be used as a classifier to distinguish early- from late-converters prior to vaccination (*Figure 3E*), with an area under the curve (AUC) of 0.825 (95% CI: 0.657–0.994) in a leave-one-out cross-validation setting. A model in which age-matched vaccinees were included from early- and late-converters returned a similar receiver operating characteristic (ROC) curve, suggesting that this signal is not age-dependent (*Figure 3F*).

While $R_{hbs}$ is able to differentiate between early- and late-converters, it seems to be worse at distinguishing non-converters. This is mainly due to a single non-converter vaccinee (H21) with a high $R_{hbs}$, signifying a high number of putative HBsAg peptide-specific TCRβ in their memory repertoire.

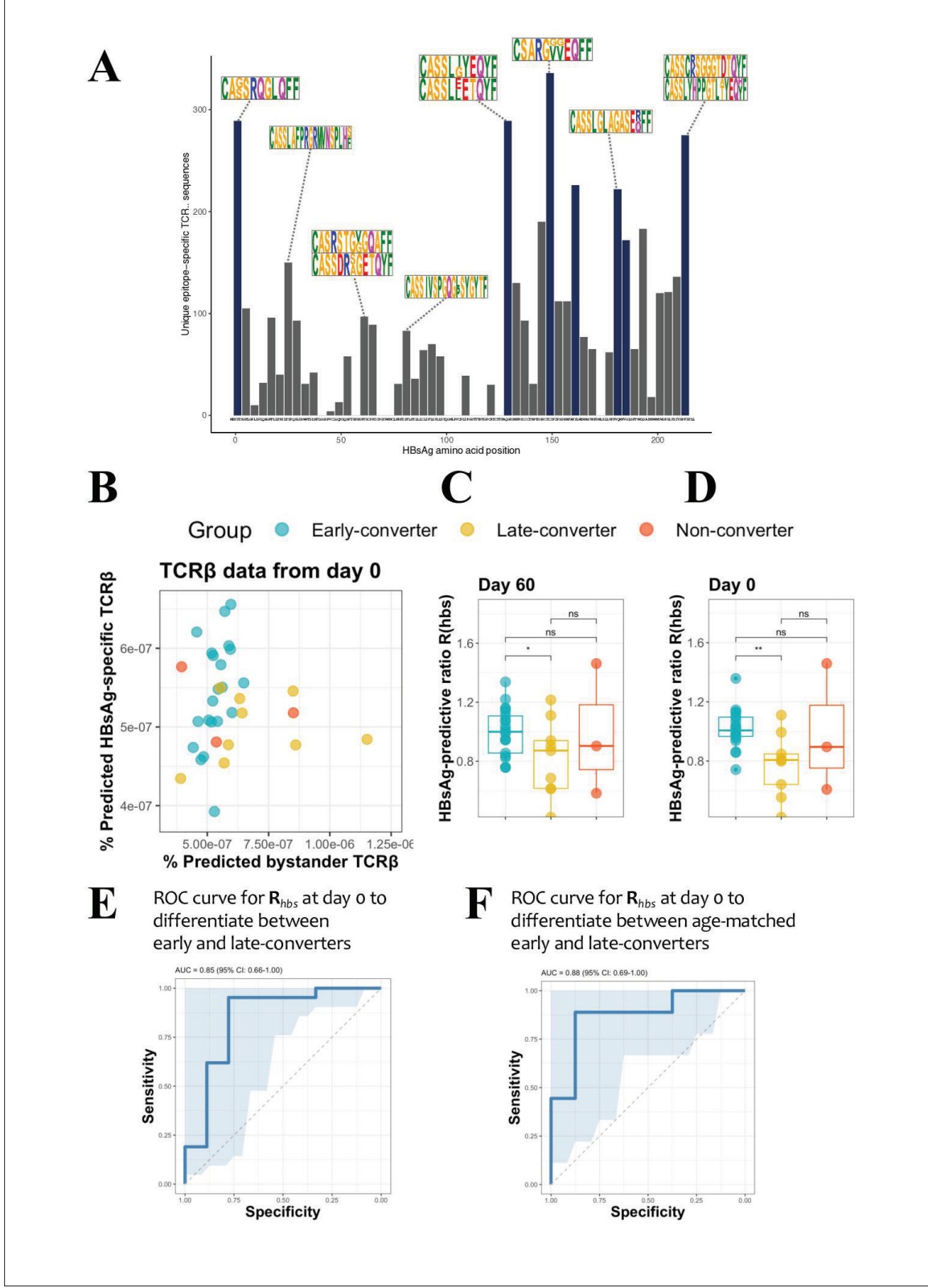

**Figure 3.** Hepatitis B surface antigen (HBsAg) peptide-specific T cell receptor β (TCRβ) identification and predictive potential of $R_{hbs}$. (**A**) Overview of the detected HBsAg epitope-specific TCRβ sequences. Each bar corresponds to unique TCRβ sequences found against a single 15mer HBsAg peptide, with 11 amino acid overlap to each subsequent peptide. Bars in blue denote those epitopes for which 10 or more volunteers had a strong T cell reaction. Motif logos on top of bars denote a sampling of the most common TCRβ amino acid sequence motifs for those epitopes. (**B**) Scatter

*Figure 3 continued on next page*

Figure 3 continued

plot with the frequency of predicted HBsAg epitope-specific and bystander TCRβ sequences at day 60. These make up respectively the numerator and denominator of the HBsAg-predictive ratio, $R_{hbs}$. Predictions done as a leave-one-out cross-validation. Each circle represents a vaccinee with the color denoting the response group (blue: early-converter, yellow: late-converter, red: non-converter). (**C**) HBsAg-predictive ratio, $R_{hbs}$, when calculated on the memory CD4 TCRβ repertoires at day 60. (**D**) HBsAg-predictive ratio, $R_{hbs}$, when calculated on the memory CD4 TCRβ repertoires at day 0. (**E**) Receiver operating characteristic (ROC) curve using $R_{hbs}$ to differentiate between early-converters and late-converters in a leave-one-out cross-validation at day 0. Reported is the area under the curve (AUC) and its 95% confidence interval. Data for B, C, D, and E are available in *Figure 3—source data 1*. (**F**) ROC curve using $R_{hbs}$ to differentiate between age-matched early-converters and late-converters in a leave-one-out cross-validation at day 0. Age-matching was accomplished retaining only samples in the age range 40–55. A Wilcoxon test was used to confirm that there was no difference in age distributions between early- and late-converters (p-value = 0.60, mean EC = 44.5 years, mean LC 45.1 years). Diagonal line denotes a random classifier. Reported is the area under the curve (AUC) and its 95% confidence interval.

The online version of this article includes the following source data and figure supplement(s) for figure 3:

**Source data 1.** Hepatitis B surface antigen (HBsAg)-predictive ratio ($R_{hb}$) data.

**Figure supplement 1.** Overview of the outcome of in vitro expansion experiments.

## Vaccine-specific conventional and regulatory memory CD4 T cells induced in early-converters

After showing evidence for the existence of vaccine-specific TCRβ sequences pre-vaccination and that individuals with a higher number of HBsAg peptide-specific clonotypes had earlier seroconversion, we attempted to link this observation to differences in vaccine-specific CD4 T cells responses using CD4 T cell assays. As $T_{REG}$ cells might suppress vaccine-induced immune responses (*Brezar et al., 2016*), we used activation markers CD40L (CD154) and 4-1BB (CD137) to help delineate the conventional ($T_{CON}$) and regulatory ($T_{REG}$) phenotypes of activated CD4 T cells. In this scheme, after 6 hr of antigen stimulation, CD40L$^+$4-1BB$^-$ can be used as a signature for antigen-specific CD4 $T_{CON}$ cells, as opposed to CD40L$^-$4-1BB$^+$ signature for antigen-specific CD4 $T_{REG}$ cells (*Elias et al., 2020*; *Schoenbrunn et al., 2012*). Additionally, we added CD25 and CD127 to better identify $T_{REG}$ cells (*Liu et al., 2006*; *Seddiki et al., 2006*) and CXCR5 to further distinguish circulatory T follicular helper cells (cT$_{FH}$) and circulatory T follicular regulatory cells (cT$_{FR}$) (*Bentebibel et al., 2011*; *Fonseca et al., 2017*) (see *Figure 4—figure supplement 1* for gating strategy).

Using the converse expression of CD40L and 4-1BB, CD40L$^+$4-1BB$^-$ and CD40L$^-$4-1BB$^+$ CD4 T cells have a CD25$^{low}$CD127$^{high}$ and CD25$^{high}$CD127$^{low}$ phenotype, respectively (*Figure 4—figure supplement 2*), and validate their use for the distinction of activated $T_{CON}$ and $T_{REG}$ cells as described before (*Schoenbrunn et al., 2012*).

We detected a significant increase in the frequency of vaccine-specific CD40L$^+$4-1BB$^-$ and CD40L$^-$4-1BB$^+$ memory CD4 T cells at day 60 in our cohort (*Figure 4A*) that correlated positively with the increase in antibody titer between day 0 and day 365 (*Figure 4B*, *Figure 4—figure supplement 3*). Upon a closer look, the induction of both signatures of vaccine-specific memory CD4 T cells was only true for early-converters (*Figure 4C*, see *Figure 4—figure supplement 4A* for non-converters and *Figure 4—figure supplement 4B* for vaccine-specific CD4 T cells). Late-converters did not show a detectable memory CD4 T cell response. Although a subset of both early- and late-converters had detectable memory CD4 T cell responses prior to vaccination, we observed no significant differences in the frequencies of CD40L$^+$4-1BB$^-$ and CD40L$^-$4-1BB$^+$ memory CD4 T cells between the two groups at day 0 (*Figure 4D*).

Collectively, flow cytometry data reveal that the expression of CD40L and 4-1BB in our ex vivo assay is consistent with our serological data and reflects the lack of seroconversion at day 60 in late-converters. However, it does not support the existence of more vaccine-specific memory CD4 T cells in early-converters prior to vaccination.

## Predictive capacity of TCRβ repertoire holds true for CD4 $T_{CON}$ immune response

Response groups used so far were established based on the dynamics of anti-HBs titers following vaccination. However, response groups can be defined based on the data of antigen specificity from the ex vivo CD4 T cell assay. Thus the frequency of CD40L$^+$4-1BB$^-$ and CD40L$^-$4-1BB$^+$ memory CD4 T cells were used to define a low and high response group at three different times points (days 60,

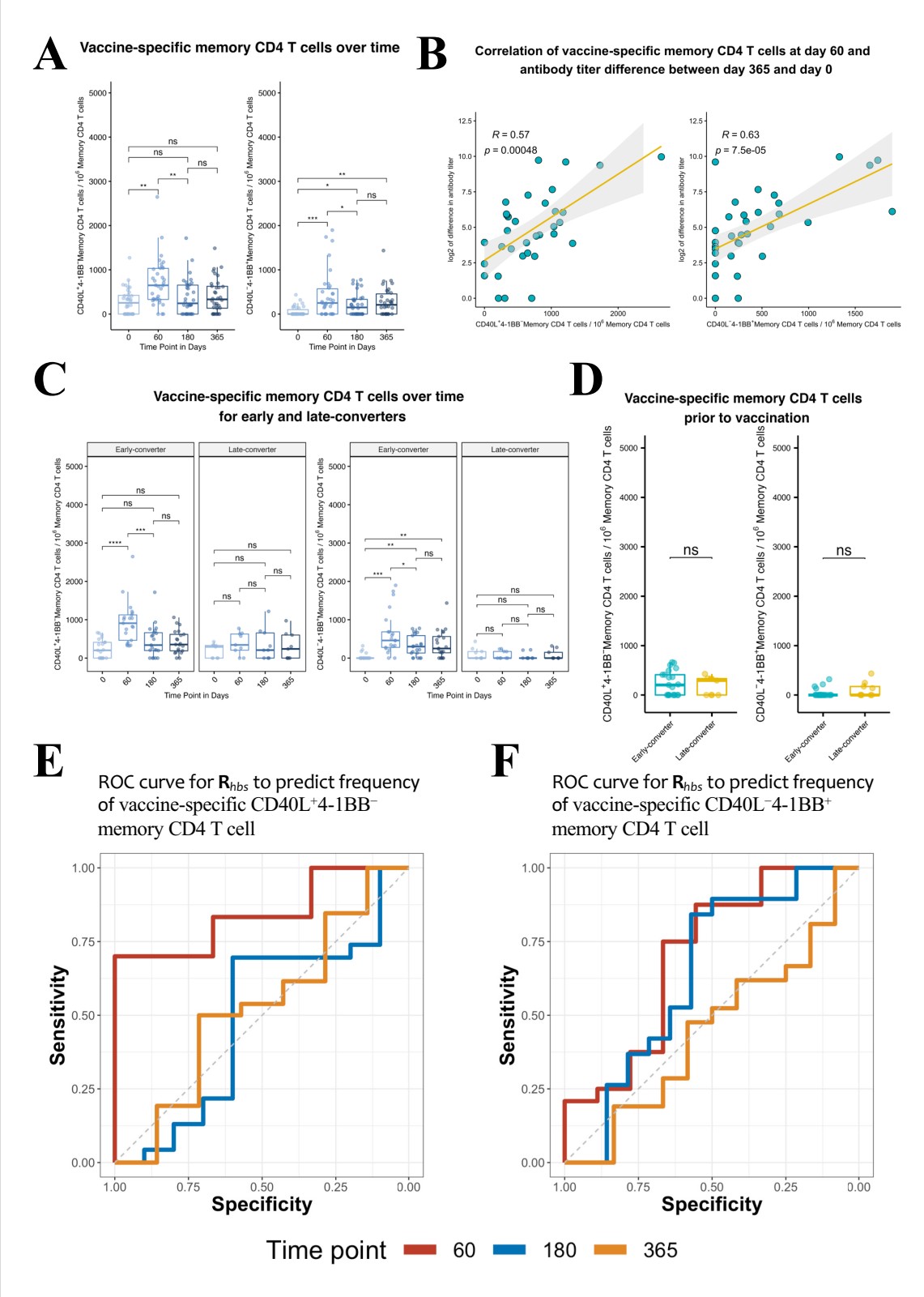

**Figure 4.** Hepatitis B vaccine induces a vaccine-specific CD40L+4-1BB− and CD40L−4-1BB+ memory CD4 T cell response in early-converter vaccinees. Peripheral blood mononuclear cells (PBMCs) from vaccinees were stimulated with 2 μg/ml of the master peptide pool (hepatitis B surface antigen [HBsAg]) and assessed for converse expression of 4-1BB and CD40L by flow cytometry on days 0, 60, 180, and 365. Shown is number of vaccine-specific memory CD4 T cells out of $10^6$ memory CD4 T cells after subtraction of responses in negative control. (**A**) Aggregate analysis from vaccinees (including

*Figure 4 continued on next page*

*Figure 4 continued*

early-, late-, and non-converters) showing a peak of vaccine-specific CD40L$^+$4-1BB$^-$ and CD40L$^-$4-1BB$^+$ memory CD4 T cell at day 60 (day 60 after first dose of the vaccine and day 30 after second dose), declining thereafter. Shown are numbers of vaccine-specific memory CD4 T cells out of 10$^6$ memory CD4 T cells. (**B**) Correlation between the difference in antibody titer between day 365 and day 0 and vaccine-specific CD40L$^+$4-1BB$^-$ and CD40L$^-$4-1BB$^+$ memory CD4 T cell at day 60. (**C**) Aggregate analysis from early- and late-converter vaccinees showing a significant induction of vaccine-specific CD40L$^+$4-1BB$^-$ and CD40L$^-$4-1BB$^+$ memory CD4 T cell in early-converters and lack thereof in late-converters. (**D**) Aggregate analysis from early- and late-converter vaccinees showing no significant differences in vaccine-specific CD40L$^+$4-1BB$^-$ and CD40L$^-$4-1BB$^+$ memory CD4 T cell at day 0. Data for A, B, C, and D are available in *Figure 4—source data 1*. (**E**) Receiver operating characteristic (ROC) curves for $R_{hbs}$ from day 0 data in a leave-one-out cross-validation compared to the frequency of vaccine-specific CD40L$^+$4-1BB$^-$ memory CD4 T cell out of 10$^6$ memory CD4 T cells for each vaccinee at time points 60 (area under the curve [AUC] = 0.84), 180 (AUC = 0.56), and 365 (AUC = 0.57). (**F**) Receiver operating characteristic (ROC) curves for $R_{hbs}$ from day 0 data in a leave-one-out cross-validation compared to the frequency of vaccine-specific CD40L$^-$4-1BB$^+$ memory CD4 T cell out of 10$^6$ memory CD4 T cells for each vaccinee at time points 60 (AUC = 0.62), 180 (AUC = 0.56), and 365 (AUC = 0.52). Statistical significance was indicated with ns p > 0.05, *p ≤ 0.05, **p ≤ 0.01, ***p ≤ 0.001, ****p ≤ 0.0001 rs, Spearman's correlation coefficient, −1≤ rs ≤ 1; rs and p-value by Spearman's correlation test.

The online version of this article includes the following source data and figure supplement(s) for figure 4:

**Source data 1.** Ex vivo T cell assay and serological data.

**Figure supplement 1.** Gating strategy of ex vivo T cell phenotyping of vaccine-specific T cells.

**Figure supplement 2.** CD40L$^+$4-1BB$^-$ and CD40L$^-$4-1BB$^+$ CD4 T cells have a T$_{CON}$ and T$_{REG}$ phenotype, respectively.

**Figure supplement 3.** Relationship between serological memory and memory CD4 T cell response to the vaccine.

**Figure supplement 4.** Hepatitis B vaccine induces a vaccine-specific CD4 T cell response in early-converter vaccinees.

180, and 365). For each time point, the same $R_{hbs}$ metric was used in a leave-one-out cross-validation as before to differentiate between these novel response group definitions. Results indicate that the $R_{hbs}$ metric is a good predictor for HBsAg-specific memory CD4 T cells with a T$_{CON}$ signature (but not T$_{REG}$ signature) identified at day 60 post-vaccination (*Figure 4E and F*). However, the classifier is less performant for the prediction of CD4 T cell response at later time points.

## An expanded subset of 4-1BB$^+$CD45RA$^-$ T$_{REG}$ cells is a prominent feature of late-converters

In order to detect any distinct signatures of early- and late-converters, we analyzed pre-vaccination flow cytometry data to examine major CD4 T cell subsets: T$_H$, T$_{REG}$, cT$_{FH}$, and cT$_{FR}$ cells.

Using manual gating in which regulatory T cells (T$_{REG}$) were defined as viable CD3$^+$CD4$^+$CD8$^-$CD25$^+$CD127$^-$CXCR5$^-$ and were further divided into CD45RA$^+$ and CD45RA$^-$ T$_{REG}$ cells, we identified a significantly higher frequency of 4-1BB$^+$ CD45RA$^-$ T$_{REG}$ cells in late-converters compared to early-converters (*Figure 5A* and *Figure 5—figure supplement 1*).

T$_{REG}$ cells showed higher 4-1BB expression compared to T$_H$, cT$_{FH}$, and cT$_{FR}$ cells (*Figure 5B*) and within T$_{REG}$ subset, CD45RA$^-$ T$_{REG}$ cells showed significantly higher expression of 4-1BB, accompanied with a higher expression of CD25, compared to CD45RA$^+$ T$_{REG}$ cells (*Figure 5C*). In this scheme, CD45RA$^-$ T$_{REG}$ can be divided into 4-1BB$^+$CD25$^{high}$ and 4-1BB$^-$CD25$^{int}$ subsets. It is worth noting here that no differences were detected in the frequency of CD45RA$^-$ or CD45RA$^+$ T$_{REG}$ cells within CD4 T cell compartment between the two groups (*Figure 5D*), and that it is the composition of T$_{REG}$ compartment that is distinct between the two groups (*Figure 5E*).

In summary, an expanded subset of 4-1BB$^+$CD45RA$^-$ T$_{REG}$ cells pre-vaccination is a prominent feature of a delayed and modest immune response to HB vaccine in our cohort.

## Discussion

In this study, we used high-throughput TCRβ repertoire profiling and ex vivo T cell assays to characterize memory CD4 T cell repertoires before and after immunization with HB vaccine, an adjuvanted subunit vaccine, and tracked vaccine-specific TCRβ clonotypes over two time points. As antigen-naïve adults were found to have an unexpected abundance of memory-phenotype CD4 T cells specific to viral antigens (*Su and Davis, 2013*; *Su et al., 2013*), we sought to investigate the influence that preexisting memory CD4 T cells can have on vaccine-induced immunity.

Commercially available HBV vaccines produce a robust and long-lasting anti-HBs response, and protection is provided by induction of an anti-HBs (antibody against HBV surface antigen) titer higher than 10 mIU/ml after a complete immunization schedule of three doses (*Meireles et al., 2015*).

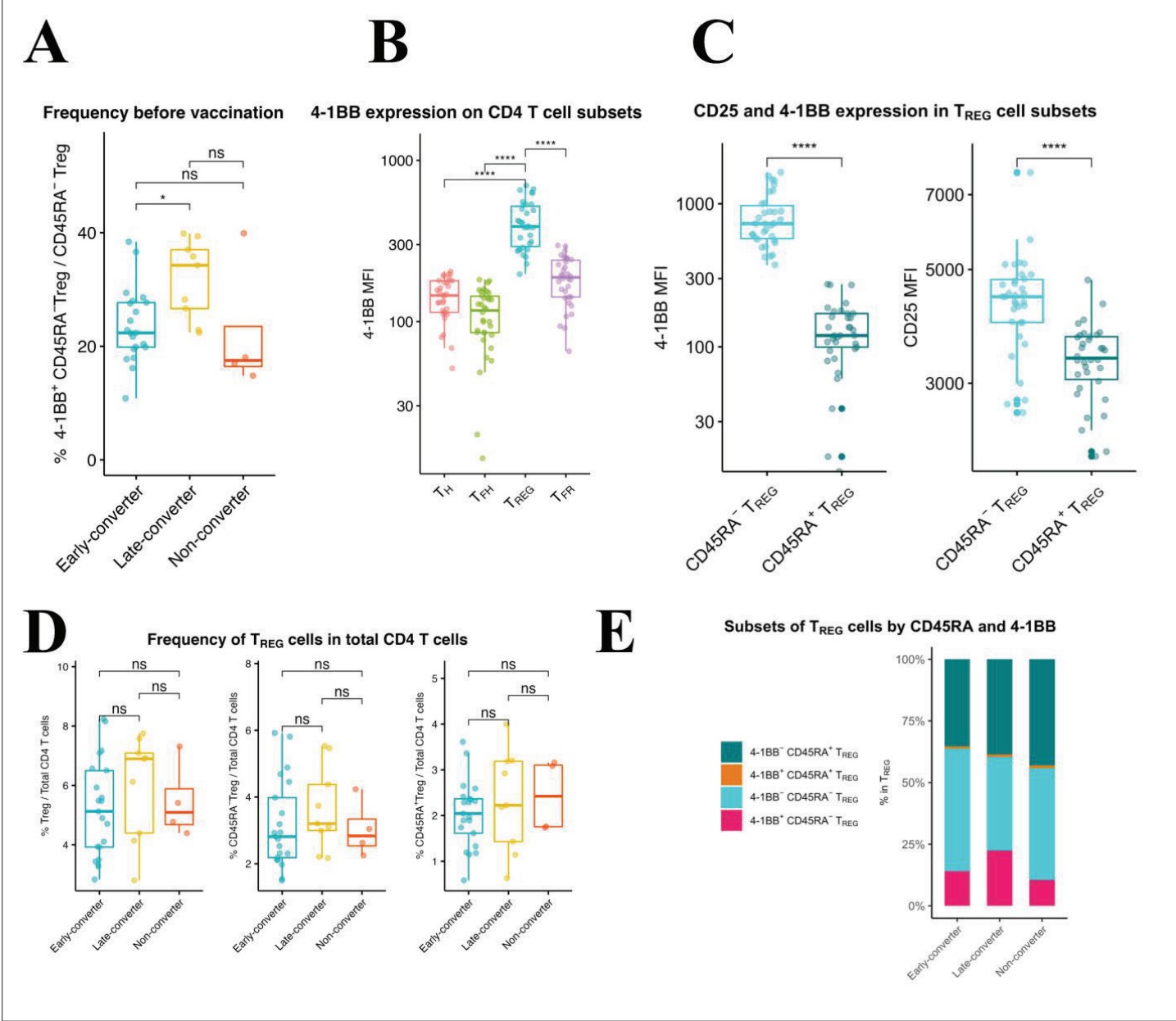

**Figure 5.** An expanded 4-1BB+CD45RA− T_REG cells within T_REG compartment is a prominent feature in late-converters prior to vaccination. Peripheral blood mononuclear cells (PBMCs) from vaccinees at day 0 (prior to vaccination) were phenotyped for expression of markers of T_REG. (**A**) Aggregate analysis of 4-1BB+CD45RA− T_REG within CD45RA− T_REG CD4 T cells in early- and late- and non-converter vaccinees before vaccination. (**B**) Aggregate analysis of the median fluorescence intensity of 4-1BB in T_H, cT_FH, T_REG, and cT_FR cells before vaccination. (**C**) Aggregate analysis of the median fluorescence intensity of 4-1BB (left panel) and CD25 (right panel) in CD45RA− T_REG and CD45RA+ T_REG cells before vaccination. Data for A, B, and C are available in *Figure 5—source data 1*. (**D**) Frequency of T_REG, CD45RA− T_REG, and CD45RA+ T_REG cells within total CD4 T cells in early-, late-, and non-converter vaccinees before vaccination. (**E**) Composition of T_REG compartment as determined by expression of 4-1BB and CD45RA in early-, late-, and non-converter vaccinees before vaccination. Statistical significance was indicated with ns p > 0.05, *p ≤ 0.05, **p ≤ 0.01, ***p ≤ 0.001, ****p ≤ 0.0001.

The online version of this article includes the following source data and figure supplement(s) for figure 5:

**Source data 1.** Frequency of 4-1BB+CD45RA− T_REG cells and median fluorescence intensity data.

**Figure supplement 1.** An expanded 4-1BB+CD45RA− T_REG cells within T_REG compartment is a prominent feature in late-converters prior to vaccination.

However, 5–10% of healthy adult vaccinees fail to produce protective titers of anti-HBs and can be classified as non-responders (*Meireles et al., 2015*). In our cohort, 13 vaccinees did not seroconvert by day 60 (30 days following administration of the second vaccine dose), as determined by antibody titer. Out of this group, nine vaccinees seroconverted by day 180 or day 365, referred to here as late-converters, and four vaccinees did not seroconvert, referred to here as non-converters.

A hallmark of adaptive immunity is a potential for memory immune responses to increase in both magnitude and quality upon repeated exposure to the antigen (*Sallusto et al., 2010*). Our systems immunology data supports the theory that preexisting memory CD4 T cell TCRβ sequences specific to HBsAg, the antigenic component of the current HB vaccine, predict which individuals will mount an early and more vigorous immune response to the vaccine as evidenced by a higher fold change in anti-HBs antibody titer and a more significant induction of antigen-specific CD4 T cells. It is postulated that preexisting memory CD4 T cell clonotypes are generated due to the highly degenerate nature of T cell recognition of antigen/MHC and are cross-reactive to environmental antigens (*Sewell, 2012*). For example, preexisting memory CD4 T cells are well-established in unexposed HIV-seronegative individuals, although at a significantly lower magnitude than HIV-exposed seronegative individuals (*Campion et al., 2014*; *Ritchie et al., 2011*), and were likely primed by exposure to environmental triggers or the human microbiome.

We and others have shown before that the TCRβ repertoire of CD4 T cells encodes the antigen exposure history of each individual and that antigen-specific TCRβ sequences could serve to automatically annotate the infection or exposure history (*DeWitt et al., 2018*; *Emerson et al., 2017*; *De Neuter et al., 2019*). In this study, we show that similar principles can be used to study vaccine responsiveness. Specifically, the recruitment of novel vaccine-specific T cell clonotypes into memory compartment following vaccination can be tracked by examining the CD4 memory TCRβ repertoire over time. While we observed no increase in the frequency of the vaccine-specific memory T cells, as the time point may have missed the peak of the clonal expansion of effector CD4 T cells as was reported before (*Blom et al., 2013*; *Kohler et al., 2012*; *Pogorelyy et al., 2018*), a significant rise in the number of unique vaccine-specific T cell clonotypes was detected. This observation is consistent with earlier studies of T cell immune repertoire that showed that antigen-specific TCRβ sequences do not always overlap with those sequences that increase in frequency after infection or vaccination (*DeWitt et al., 2015*). More interestingly, individuals with the earlier and more robust response against the vaccine had a telltale antigen-specific signature in their memory TCRβ repertoire prior to vaccination, despite the lack of HBsAg antibodies and prior vaccination history.

Detection of this vaccine-specific signature was possible due to the development of a novel predictive model that used epitope-specific TCRβ sequences from one set of individuals to make predictions about another. This is a non-trivial task as the MHC class II molecules vary between vaccinees, which in turn allows for great variation of the presented epitopes. However, we were able to cover the entire HBsAg and obtained a sufficient sample size to likely cover the most immunoprevalent epitopes. It can be presumed that we do not capture the full HBsAg T cell response, but obtain enough sequences which are representative for the response as a whole. In addition, a correction factor was needed to account for the occurrence of bystander-activated T cells within the original epitope-specific TCRβ sequences. Indeed, in those vaccinees without an early seroconversion (at day 60), putative vaccine-specific T cells might be induced due to bystander activation. This was supported by predictions using the TCRex tool (*Gielis et al., 2019*), which matched these TCR sequences to common viral or other epitopes. It is of note that these TCRβ sequences are matched with CD8 T cell epitopes, while they originate from isolated CD4 T cells. This is likely due to the great similarity between the TCRβ sequences of CD4 and CD8 T cells as noted in prior research (*De Neuter et al., 2018*).

However, our in vitro antigen-specific data, using an assay that enables discrimination of $T_{CON}$ and $T_{REG}$ cells using the converse expression of the activation markers CD40L and 4-1BB (*Frentsch et al., 2005*; *Schoenbrunn et al., 2012*), failed to show a significant difference in preexisting antigen-specific CD4 T cells between early- and late-converters prior to vaccine administration. It is plausible that our activation proteins, CD40L and 4-1BB, might be unsuitable to detect preexisting memory CD4 T cells but this is unlikely as both proteins were used successfully in similar studies (*Bacher et al., 2014a*; *Bacher et al., 2014b*).

Another plausible explanation is that the signal is below the detection limit of the assay and that more sensitive techniques that require pre-enrichment of CD40L$^+$ and 4-1BB$^+$ T cells (using magnetic

beads) (*Bacher et al., 2013*) or cultured ELISpot assay (*Reece et al., 2004*) are needed to capture preexisting vaccine-specific memory CD4 T cells directly from human peripheral blood. A similar disagreement between B cell receptor clonotype diversity underlying vaccine-induced response and a conventional B cell ELISpot assay was reported before following HB vaccination (*Galson et al., 2016*).

T$_{REG}$ cells represent about 5–10% of human CD4 T cell compartment and are identified by the constitutive surface expression of CD25, also known as IL-2 receptor α subunit (IL-2Ra), and the nuclear expression of forkhead family transcription factor 3 (Foxp3), a lineage specification factor of T$_{REG}$ cells (*Rudensky, 2011*). Regulatory memory T cells play a role in the mitigation of tissue damage induced by effector memory T cells during protective immune responses, resulting in a selective advantage against pathogen-induced immunopathology (*Garner-Spitzer et al., 2013*; *Lanteri et al., 2009*; *Lin et al., 2018*; *Lovelace and Maecker, 2018*). Several studies have identified CD4 T$_{REG}$ cells with specificity to pathogen-derived peptides in murine models and showed evidence for an induced expansion of T$_{REG}$ cells followed by an emergence and a long-term persistence of T$_{REG}$ cells with a memory phenotype and potent immunosuppressive properties (*Lin et al., 2018*; *Sanchez et al., 2012*). Blom et al. reported a significant and transient activation of T$_{REG}$ cells (identified by upregulation of CD38 and Ki67) in humans 10 days after administration of live attenuated yellow fever virus 17D vaccine (*Blom et al., 2013*). The induction of vaccine-specific T$_{REG}$ cells in our cohort is unexpected and the role it might play in vaccine-induced immunity warrants further investigation.

The association of an expanded 4-1BB$^+$ CD45RA$^-$ T$_{REG}$ subset with a delayed immune response to HB vaccine was not described before. Miyara et al. showed that blood contains two distinct subsets of stable and suppressive T$_{REG}$ cells: resting T$_{REG}$, identified as FOXP3$^{low}$CD45RA$^+$ CD4 T cells, and activated T$_{REG}$, identified as FOXP3$^{high}$CD45RA$^-$ CD4 T cells. They further noted that activated T$_{REG}$ cells constitute a minority subset within cord blood T$_{REG}$ cells and increase gradually with age (*Miyara et al., 2009*). As activated T$_{REG}$ cells were shown to have an increased expression of proteins indicative of activation, including ICOS and HLA-DR (*Booth et al., 2010*; *Ito et al., 2008*; *Mason et al., 2015*; *Miyara et al., 2009*; *Mohr et al., 2018*), it might be the case that an upregulation of 4-1BB is one more feature of this population or a subset thereof. Moreover, T$_{REG}$ cells in mice were shown to modulate T$_{FH}$ formation and germinal center (GC) B cell responses and to diminish antibody production in a CTLA-4 mediated suppression (*Wing et al., 2014*). Interestingly, CD45RA$^-$ T$_{REG}$ cells were shown to be more rich in preformed CTLA-4 stored in intracellular vesicles compared to CD45RA$^+$ T$_{REG}$ cells (*Miyara et al., 2009*).

4-1BB was shown to be constitutively expressed by T$_{REG}$ cells (*McHugh et al., 2002*) and that 4-1BB$^+$ T$_{REG}$ cells are functionally superior to 4-1BB$^-$ T$_{REG}$ cells in both contact-dependent and contact-independent immunosuppression (*Kachapati et al., 2012*). 4-1BB$^+$ T$_{REG}$ cells are the major producers of the alternatively spliced and soluble isoform of 4-1BB among T cells (*Kachapati et al., 2012*). 4-1BB was shown before to be preferentially expressed on T$_{REG}$ cells compared with other non-regulatory CD4 T cell subsets (*McHugh et al., 2002*) and that 4-1BB-costimualtion induces the expansion of T$_{REG}$ cells both in vitro and in vivo (*Zheng et al., 2004*). Moreover, agonistic anti-4-1BB mAbs have been shown to abrogate T cell-dependent antibody responses in vivo (*Mittler et al., 1999*) and to ameliorate experimental autoimmune encephalomyelitis by skewing the balance against T$_H$17 differentiation in favor of T$_{REG}$ differentiation (*Kim et al., 2011*). It is plausible that the expansion 4-1BB$^+$CD45RA$^-$ T$_{REG}$ cells in late-converters is involved in the suppression of GC vaccine-specific T$_{FH}$ cells and the ensuing antibody response in our cohort, but this remains speculative and further research is warranted.

It is enticing to speculate that the preexisting memory CD4 T cells result from the complex interplay between cellular immunity and the human microbiome. A role for the microbiota in modulating immunity to viral infection was suggested in 1960s (*Robinson and Pfeiffer, 2014*), and since then we gained better understanding of the impact of the various components of the microbiota including bacteria, fungi, protozoa, archaea, and viruses on the murine and human immune systems (*Winkler and Thackray, 2019*).

Viral clearance of HB virus infection depends on the age of exposure and neonates and young children are less likely to spontaneously clear the virus (*Yuen et al., 2018*). Han-Hsuan et al. have shown evidence in mice that this age dependency is mediated by gut microbiota that prepare the liver immunity system to clear HBV, possibly via a TLR4 signaling pathway (*Chou et al., 2015*). In this study, young mice that have not reached an equilibrium in the gut microbiota exhibited prolonged HBsAg persistence, impaired anti-HBs antibody production, and limited Hepatitis B core antigen

(HBcAg)-specific IFNγ⁺ splenocytes. More recently, Tingxin et al. provided evidence for a critical role of the commensal microbiota in supporting the differentiation of GC B cells, through follicular T helper (T$_{FH}$) cells, to promote the anti-HBV humoral immunity (**Wu et al., 2019**).

Our study bears some intrinsic limitations. A major drawback is the restricted number of days at which TCRβ repertoire was profiled, as vaccine-specific perturbations within the repertoire may occur at different time points for early-, late-, and non-converters. Additionally, more in-depth characterization and functional studies on 4-1BB⁺CD45RA⁻ T$_{REG}$ cells could have helped shed more light on the role they play in vaccine-induced immunity. Future studies in larger cohorts and with a more comprehensive TCRβ repertoire profiling and CD4 T cells immunophenotyping are required to validate our findings.

In conclusion, our analysis of the memory CD4 T cell repertoire has uncovered a role for preexisting memory CD4 T cells in naïve individuals in mounting an earlier and more vigorous immune response to HB vaccine and argue for the utility of pre-vaccination TCRβ repertoire in the prediction of vaccine-induced immunity. Moreover, we identify a subset of 4-1BB⁺ memory T$_{REG}$ cells that is expanded in individuals with delayed immune response to the vaccine, which might further explain the heterogeneity of response to HB vaccine.

# Materials and methods

## Key resources table

| Reagent type (species) or resource | Designation | Source or reference | Identifiers | Additional information |
|---|---|---|---|---|
| Antibody | CD3-BV510 (SK7) (mouse monoclonal) | BioLegend | Cat# 344828 | FACS (1/20 per test) |
| Antibody | CD3-PerCP (BW264/56) (mouse monoclonal) | Miltenyi Biotec | Cat# 130-113-131 | FACS (1/50 per test) |
| Antibody | CD4-APC (REA623) (recombinant antibodies, REAfinity) | Miltenyi Biotec | Cat# 130-113-222 | FACS (1/50 per test) |
| Antibody | CD4-APC (RPA-T4) (mouse monoclonal) | BD Biosciences | Cat# 555349 | FACS (1/5 per test) |
| Antibody | CD4-PerCP/Cy5.5 (RPA-T4) (mouse monoclonal) | BioLegend | Cat# 300530 | FACS (1/20 per test) |
| Antibody | CD8-VioGreen (REA734) (recombinant antibodies, REAfinity) | Miltenyi Biotec | Cat# 130-110-684 | FACS (1/50 per test) |
| Antibody | CD8-Pacific Orange (3B5) (mouse monoclonal) | Invitrogen | Cat# MHCD0830 | FACS (1/20 per test) |
| Antibody | CD8-APC/Cy7 (SK1) (mouse monoclonal) | BioLegend | Cat# 344714 | FACS (1/20 per test) |
| Antibody | CD40L-PE (5C8) (mouse monoclonal) | Miltenyi Biotec | Cat# 130-092-289 | FACS (1/10 per test) |
| Antibody | CXCR5-PE-Cy7 (MU5UBEE) (mouse monoclonal) | eBioscience | Cat# 25-9185-42 | FACS (7/100 per test) |
| Antibody | CD45RA-AF488 (HI100) (mouse monoclonal) | BioLegend | Cat# 304114 | FACS (1/20 per test) |
| Antibody | CD45RO-PE (UCHT1) (mouse monoclonal) | BD Biosciences | Cat# 555493 | FACS (1/5 per test) |
| Antibody | CD25-BV421 (M-A251) (mouse monoclonal) | BioLegend | Cat# 356114 | FACS (1/20 per test) |
| Antibody | CD127-BV785 (A019D5) (mouse monoclonal) | BioLegend | Cat# 351330 | FACS (1/20 per test) |
| Antibody | CD137-PE (4B4-1) (mouse monoclonal) | BioLegend | Cat# 309804 | FACS (1/20 per test) |
| Antibody | CD154-APC (5C8) (mouse monoclonal) | Miltenyi Biotec | Cat# 130-113-603 | FACS (3/100 per test) |
| Antibody | Anti-CD40 (HB14) (mouse monoclonal) | Miltenyi Biotec | Cat# 130-094-133 | Assay (1 µg/ml) |
| Antibody | Anti-CD28 (CD28.2) (mouse monoclonal) | BD Biosciences | Cat# 556620 | Assay (1 µg/ml) |
| Peptide, recombinant protein | Single peptides of HBsAg | JPT Peptide Technologies | Customized | 54 single peptides |
| Peptide, recombinant protein | Peptide matrix pools of HBsAg | JPT Peptide Technologies | Customized | A set of 15 matrix pools each with 7–8 peptides |
| Peptide, recombinant protein | Master peptide pool of HBsAg | JPT Peptide Technologies | Customized | A pool of 54 single peptides |

*Continued on next page*

*Continued*

| Reagent type (species) or resource | Designation | Source or reference | Identifiers | Additional information |
|---|---|---|---|---|
| Commercial assay or kit | CD4 MicroBeads | Miltenyi Biotec | Cat# 130-045-101 | |
| Commercial assay or kit | DNA/RNA Shield | Zymo Research | Cat# R1100-50 | |
| Commercial assay or kit | Quick-DNA Microprep kit | Zymo Research | Cat# D3020 | |
| Commercial assay or kit | ImmunoSEQ hsTCRB sequencing kit | Adaptive Biotechnologies | | |
| Commercial assay or kit | Quick-RNA Microprep kit | Zymo Research | Cat# R1050 | |
| Commercial assay or kit | QIAseq Immune Repertoire RNA Library kit | Qiagen | Cat# 333705 | |
| Commercial assay or kit | Qubit dsDNA HS Assay | Thermo Fisher Scientific | Cat# Q32854 | |
| Software, algorithm | FlowJo version 10.5.3 | Tree Star | RRID:SCR_002865 | |
| Software, algorithm | immunoSEQ analyzer (v2) | Adaptive Biotechnologies | | |
| Software, algorithm | TCRex web-based | (*Gielis et al., 2019*), https://tcrex.biodatamining.be/ | | |
| Software, algorithm | R | https://www.r-project.org | RRID:SCR_001905 | |
| Software, algorithm | RStudio | http://www.rstudio.com/ | RRID:SCR_000432 | |
| Software, algorithm | ggplot2 (V3.3.2) | (*Wickham, 2016*) https://ggplot2.tidyverse.org/ | RRID:SCR_014601 | |
| Software, algorithm | ggpubr (V0.2.5) | https://CRAN.R-project.org/package=ggpubr | RRID:SCR_021139 | |
| Software, algorithm | rstatix (0.7.0) | https://CRAN.R-project.org/package=rstatix | RRID:SCR_021240 | |
| Other | Sytox blue | Invitrogen | Cat# S34857 | FACS (1/500 per test) |
| Other | Fixable viability dye Zombie NIR | BioLegend | Cat# 423106 | FACS (1/50 per test) |
| Other | Carboxyfluorescein succinimidyl ester (CFSE) | Invitrogen | Cat# C34554 | 2 µM staining solution |
| Other | Engerix-B | GlaxoSmithKline | | 20 µg |
| Other | Varicella zoster virus (VZV) lysate | Microbix Biosystem | Cat# EL-03–02 | Viral lysate, assay (4 µg/ml) |

## Human study design and clinical samples

A total of 34 healthy individuals (20-29 years: 10, 30-39 years: 7, 40-49 years: 16, 50+ years: 1) without a history of HBV infection or previous HB vaccination were recruited in this study after obtaining written informed consent. Individuals were vaccinated with an HB vaccine by intramuscular (m. deltoideus) injection (Engerix-B containing 20 µg dose of alum-adjuvanted HBsAg, GlaxoSmithKline) on days 0 and 30 (and on day 365). At days 0 (pre-vaccination), 60, 180, and 365 (*Figure 1A*), peripheral blood samples were collected on spray-coated lithium heparin tubes, spray-coated K2EDTA (dipotassium ethylenediamine tetra-acetic acid) tubes, and serum tubes (Becton Dickinson, NJ).

## Peripheral blood mononuclear cells

Peripheral blood mononuclear cells (PBMCs) were isolated by Ficoll-Paque Plus gradient separation (GE Healthcare, Chicago, IL). Cells were stored in 10% dimethyl sulfoxide in fetal bovine serum (Thermo Fisher Scientific, Waltham, MA). After thawing and washing cryopreserved PBMC, cells were cultured in AIM-V medium that contained L-glutamine, streptomycin sulfate at 50 µg/ml, and gentamicin sulfate at 10 µg/ml (Thermo Fisher Scientific, Waltham, MA) and supplemented with 5% human serum (One Lambda, Canoga Park, CA).

## Serology and complete blood count

Serum was separated and stored immediately at – 80°C until time of analysis. Anti-HBs antibody was titrated in serum from day 0, 60, 180, and 365 using Roche Elecsys Anti-HBs antibody assay on an

Elecsys 2010 analyzer (Roche, Basel, Switzerland). An anti-HBs titer above 10 IU/ml was considered protective (*Keating and Noble, 2003*).

Serum IgG antibodies to CMV, EBV viral-capsid antigen, and HSV-1 and -2 were determined using commercially available sandwich ELISA kits in accordance with the manufacturer's instructions.

A complete blood count including leukocyte differential was run on a hematology analyzer (ABX MICROS 60, Horiba, Kyoto, Japan).

## Sorting of memory CD4 T cells

Total CD4 T cells were isolated by positive selection using CD4 magnetic microbeads (Miltenyi Biotech, Bergisch Gladbach, Germany). Memory CD4 T cells were sorted after gating on single viable CD3+CD4+CD8−CD45RO+ cells. The following fluorochrome-labeled monoclonal antibodies were used for staining: CD3-PerCP (BW264/56) (Miltenyi Biotech), CD4-APC (RPA-T4), and CD45RO-PE (UCHT1) (both from Becton Dickinson, Franklin Lakes, NJ) and CD8-Pacific Orange (3B5) (from Thermo Fisher Scientific, Waltham, MA). Cells were stained at room temperature for 20 min and sorted with FACSAria II (Becton Dickinson, Franklin Lakes, NJ). Sytox blue (Thermo Fisher Scientific, Waltham, MA) was used to exclude non-viable cells.

## Single peptides, peptide matrix pools, and epitope mapping

A set of 15-mers peptides with an 11-amino acid overlap spanning the 226 amino acids along the small S protein of HBsAg, also designated as small HBs (*Shouval, 2003*), were synthesized by JPT Peptide Technologies (Berlin, Germany). The set, composed of 54 **single peptides** (see *Supplementary file 1*), was used in a matrix-based strategy to map epitopes against which the immune response is directed as described before (*Precopio et al., 2008*). The matrix layout enables efficient identification of epitopes within the antigen using a minimal number of cells. For this purpose, a matrix of 15 pools, 7 rows, and 8 columns, referred to as **peptide matrix**, was designed so that each peptide is in exactly one row-pool and one column-pool, thereby allowing for the identification of positive peptides at the intersection of positive pools. Matrix pools that induced a CD4 T cell response (as determined by CD40L/CD154 assay described below) which meets the threshold criteria for a positive response were considered in the deconvolution process. Top six single peptides were considered for peptide-specific T cell expansion and sorting. A **master peptide pool** is composed of all of the 54 single peptides and was used to identify and sort total vaccine-specific CD4 T cells. Each peptide was used at a final concentration of 2 µg/ml. VZV lysate was purchased from Microbix Biosystem (Mississauga, Canada).

## Ex vivo T cell stimulation (CD40L/CD154 assay)

Thawed PBMCs from each vaccinee were cultured in AIM-V medium that contained L-glutamine, streptomycin sulfate at 50 µg/ml, and gentamicin sulfate at 10 µg/ml (GIBCO, Grand Island, NY) and supplemented with 5% human serum (One Lambda, Canoga Park, CA). Cells were stimulated for 6 hr with 2 µg/ml of each of the 15 peptide matrix pools in the presence of 1 µg/ml anti-CD40 antibody (HB14) (purchased from Miltenyi Biotec, Bergisch Gladbach, Germany) and 1 µg/ml anti-CD28 antibody (CD28.2) (purchased from BD Biosciences, Franklin Lakes, NJ).

Cells were stained using the following fluorochrome-labeled monoclonal antibodies: CD3-PerCP (BW264/56), CD4-APC (REA623), CD8-VioGreen (REA734), and CD40L-PE (5C8) (purchased from Miltenyi Biotec, Bergisch Gladbach, Germany). Viability dye Sytox blue from Invitrogen (Thermo Fisher Scientific, Waltham, MA) was used to exclude non-viable cells. Data was acquired on FACSAria II using Diva Software, both from BD Biosciences (Franklin Lakes, NJ), and analyzed on FlowJo software version 10.5.3 (Tree Star, Inc, Ashland, OR). Fluorescence-minus-one controls were performed in pilot studies. Gates for CD40L+CD4 T cells were set using cells left unstimulated.

## In vitro T cell expansion and cell sorting

Thawed PBMCs were labeled with CFSE (Invitrogen, Carlsbad, CA) and cultured in AIM-V medium that contained L-glutamine, streptomycin sulfate at 50 µg/ml, and gentamicin sulfate at 10 µg/ml (GIBCO, Grand Island, NY) and supplemented with 5% human serum (One Lambda, Canoga Park, CA). Cells were stimulated for 7 days with 2 µg/ml of selected single peptides in addition to the master peptides pool. Cells were also stimulated with 4 µg/ml of a VZV lysate and used as a baseline control. Cells were

stained using the following fluorochrome-labeled monoclonal antibodies: CD3-PerCP (BW264/56), CD4-APC (REA623), and CD8-VioGreen (REA734) (purchased from Miltenyi Biotec, Bergisch Gladbach, Germany). Viability dye Sytox blue from Invitrogen (Thermo Fisher Scientific, Waltham, MA) was used to exclude non-viable cells. Single viable CFSE$^{low}$ CD3$^+$ CD8$^-$ CD4$^+$ T cells were sorted into 96-well PCR plates containing DNA/RNA Shield (Zymo Research, Irvine, CA) using FACSAria II and Diva Software (BD Biosciences, Franklin Lakes, NJ). For each of the selected single peptides, 500 cells were sorted in two technical replicates. For the master peptide pool and VZV lysate, 1000 cells were sorted in two technical replicates. Plates were immediately centrifuged and kept at –20°C before TCR cDNA library preparation and sequencing.

## TCRβ cDNA library preparation and sequencing of memory CD4 T cells

DNA was extracted from sorted memory CD4 T cells using Quick-DNA Microprep kit (Zymo Research, Irvine, CA). ImmunoSEQ hsTCRB sequencing kit (Adaptive Biotechnologies, Seattle, WA) was used to profile TCRβ repertoire following the manufacturer's protocol.

After quality control using Fragment Analyzer (Agilent, Santa Clara, CA), libraries were pooled with equal volumes. The concentration of the final pool was measured with the Qubit dsDNA HS Assay kit (Thermo Fisher Scientific, Waltham, MA). The final pool was processed to be sequenced on the Miseq and NextSeq platforms (Illumina, San Diego, CA). Memory CD4 T cells of one of the vaccinees (H42, a non-converter) was not sequenced due to a capacity issue.

## TCR cDNA library preparation and sequencing of CFSE$^{low}$ CD4 T cells

RNA was extracted from each of the two technical replicates of sorted CFSE$^{low}$ CD4 T cells using Quick-RNA Microprep kit (Zymo Research, Irvine, CA). Without measuring the resulting RNA concentration, an RNA-based library preparation was used. The QIAseq Immune Repertoire RNA Library kit (Qiagen, Venlo, The Netherlands) amplifies TCRα, -β, -γ, and -δ chains. After quality control using Fragment Analyzer (Agilent, Santa Clara, CA), concentration was measured with the Qubit dsDNA HS Assay kit (Thermo Fisher Scientific, Waltham, MA) and pools were equimolarly pooled and prepared for sequencing on the NextSeq platform (Illumina, San Diego, CA).

## TCRβ sequence analysis

TCRβ clonotypes were identified as previously described (*De Neuter et al., 2019*) where a unique TCRβ clonotype is defined as a unique combination of a V gene, CDR3 amino acid sequence, and J gene. All memory CD4 T cell DNA-based TCRβ sequencing reads were annotated using the immunoSEQ analyzer (v2) from Adaptive Biotechnologies. All small bulk RNA-based TCR sequencing reads were annotated using the MiXCR tool (v3.0.7) from the FASTQ files. As all RNA-based TCR sequencing experiments featured two technical replicates, only those TCR sequences that occurred in both replicates were retained and their counts were summed. Tracking of vaccine-specific TCRβ clonotypes is based on exact TCRβ CDR3 amino acid matches to remove any bias introduced by the different VDJ annotation pipelines. Non-HBsAg TCR annotations were done with the TCRex web tool (*Gielis et al., 2019*) on the July 24, 2019, using version 0.3.0. Breadth of the TCR sequences is defined as the number of unique TCR clonotypes belonging to a given set, divided by the total number of unique clonotypes in a repertoire. Inference of similar epitope binding between two TCR sequences is defined according to the Hamming distance ($d$) calculated on the CDR3 amino acid sequence with a cutoff $c = 1$, as supported by prior research where it was shown that it was equivalent to the performance of more complex methods such as TCRdist (*Dash et al., 2017*) or k-mer clustering when applied to TCRβ chains only (*De Neuter et al., 2018*; *Valkiers et al., 2021*). All scripts used in this analysis are available via GitHub (https://github.com/pmeysman/HepBTCR, copy archived at swh:1:rev:bdda21dde671ac2e424e85bd270efafa719d4cb4, *Meysman, 2022*).

## Predictive HBs-response model

From the single peptide data generated in the epitope mapping experiments, we aimed to create a predictive model to enumerate the HBs response from full TCRβ repertoire data. This approach allows for predictions that are epitope-specific rather than simply vaccine-specific. This model was applied in a leave-one-out cross-validation so that vaccine-specific TCRβ sequences from a vaccinee are not used to make predictions for the same vaccinee. While the predictive model is derived from

epitope-specific data, it cannot be guaranteed that some of the expanded CD4 T cells detected in the in vitro assay are not due to bystander activation. Vaccine-specific TCRβ sequences of vaccinees who did not respond to the vaccine at day 60 (late-converters and non-converters) are expected to be more enriched in cells triggered to expand due to bystander activation. Indeed, running the set of vaccine-specific TCRβ sequences through the TCRex webtool (*Gielis et al., 2019*) reveals that several TCRs are predicted to be highly similar to those reactive to the CMV NLVPMVATV epitope (enrichment p-value < 0.001 when compared to the TCRex background repertoire) and the Mart-1 variant ELAGIGILTV epitope (p-value < 0.001), which supports the notion that some of these TCRβ sequences might not be specific to HBsAg. This set of vaccine-specific TCRβ sequences can thus be used to make predictions about possible TCRβ sequences due to bystander activation of CD4 T cells, that is, common TCRβ sequences that might be present as false positives. The final output of the model is thus a ratio, $R_{hbs}$, for any repertoire $rep_i$ describing a set of TCRβ sequences $t_{repi}$:

$$R_{hbs}\left(t_{repi}\right) = \frac{\sum_{pep=1}^{54}\left|\left\{x \in t_{repi}|min_{y \in t_{pep}}\ d\left(x,y\right)<c\right\}\right|/|t_{pep}|}{\left|\left\{x \in t_{repi}|min_{y \in t_{bystander}}\ d\left(x,y\right)<c\right\}\right|/|t_{bystander}|}$$

with $t_{pep}$ as the set of TCRβ sequences occurring in both biological replicates for a single sample and a single peptide (*pep*) from the HBsAg epitope mapping experiment, and $t_{bystander}$ as the set of TCRβ sequences occurring in both biological replicates of the master peptide pool in any of the non-responding samples. Thus, the ratio signifies the number of TCR clonotypes predicted to be reactive against one of the HBsAg peptides, normalized by a count of putative false positive predictions from bystander T cells.

## Ex vivo T cell phenotyping of vaccine-specific T cells

Thawed PBMCs from each vaccinee were cultured in AIM-V medium that contained L-glutamine, streptomycin sulfate at 50 μg/ml, and gentamicin sulfate at 10 μg/ml (GIBCO, Grand Island, NY) and supplemented with 5% human serum (One Lambda, Canoga Park, CA). Cells were stimulated for 6 hr with 2 μg/ml of a master peptide pool representing the full length of the small surface envelope protein of HB, in the presence of 1 μg/ml anti-CD40 antibody (HB14) (purchased from Miltenyi Biotec, Bergisch Gladbach, Germany) and 1 μg/ml anti-CD28 antibody (CD28.2) (purchased from BD Biosciences, Franklin Lakes, NJ). Cells were stained using the following fluorochrome-labeled monoclonal antibodies: CD3-BV510 (SK7), CD4-PerCP/Cy5.5 (RPA-T4), CD8-APC/Cy7 (SK1), CD45RA-AF488 (HI100), CD25-BV421 (M-A251), CD127-BV785 (A019D5), and CD137-PE (4-1BB) (purchased from BioLegend, San Diego, CA), CXCR5 (CD185)-PE-eFluor 610 (MU5UBEE) (from eBioscience, Thermo Fisher Scientific, Waltham, MA) and CD40L-APC (5C8) (purchased from Miltenyi Biotec, Bergisch Gladbach, Germany). Fixable viability dye Zombie NIR from BioLegend (San Diego, CA) was used to exclude non-viable cells. Data was acquired on FACSAria II using Diva Software, both from BD Biosciences (Franklin Lakes, NJ), and analyzed on FlowJo software version 10.5.3 (Tree Star, Inc, Ashland, OR) using gating strategy shown in *Figure 4—figure supplement 1A*. Fluorescence-minus-one controls were performed in pilot studies. Gates for CD40L+ and 4-1BB+ CD4 T cells (*Figure 4—figure supplement 1B*) were set using cells left unstimulated (negative control contained DMSO at the same concentration used to solve peptide pools). In order to account for background expression of CD40L and 4-1BB on CD4 T cells, responses in cells left unstimulated were subtracted from the responses to peptides, and when peptides-specific CD40L+ or 4-1BB+ CD4 T cells were not significantly higher than those detected for cells left unstimulated (using one-sided Fisher's exact test), values were mutated to zero.

## Statistics and data visualization

The two-sided Fisher's exact test was used to evaluate the significance of relationship between early/late-converters and CMV, EBV, or HSV seropositivity. For the visualization of marker expression, TCRβ counts and cell frequencies between time points or groups of vaccinees, ggplot2 (v3.3.2), ggpubr (v0.2.5), and rstatix (v0.7.0) packages in R were used. The Wilcoxon rank sum test and the Wilcoxon signed-rank test were used to compare two or more groups, with unpaired and paired analysis as necessary. Bonferroni correction was applied when multiple comparisons were made. The nonparametric Spearman's rank-order correlation was used to test for correlation. We used the following

convention for symbols indicating statistical significance; ns p > 0.05, *p ≤ 0.05, **p ≤ 0.01, ***p ≤ 0.001, ****p ≤ 0.0001.

## Acknowledgements

We thank all the volunteers in the study, and all the nurses, lab technicians, researchers, and staff in the clinical biology laboratory of Antwerp University Hospital and the Centre for the Evaluation of Vaccination at the Vaccine and Infectious Disease Institute that were involved in the study. In addition, we wish to thank Nick de Vrij and Sebastiaan Valkiers for critically reading the manuscript for clarity. We further thank the reviewers and editors for their many suggestions that helped improve this work.

## Additional information

### Competing interests

Nina Keersmaekers: Viggo Van Tendeloo: The other authors declare that no competing interests exist.

### Funding

| Funder | Grant reference number | Author |
| --- | --- | --- |
| Universiteit Antwerpen | | George Elias<br>Esther Bartholomeus<br>Nicolas De Neuter |
| Research Foundation Flanders | | Pieter Meysman<br>Kris Laukens<br>Benson Ogunjimi |
| American Lebanese Syrian Associated Charities | | Aisha Souquette<br>Paul G Thomas |
| National Institute of Allergy and Infectious Diseases | | Aisha Souquette<br>Paul G Thomas |

The funders had no role in study design, data collection and interpretation, or the decision to submit the work for publication.

### Author contributions

George Elias, Formal analysis, Investigation, Methodology, Software, Visualization, Writing – original draft; Pieter Meysman, Conceptualization, Data curation, Formal analysis, Supervision, Visualization, Writing – review and editing; Esther Bartholomeus, Data curation, Formal analysis, Investigation, Methodology, Writing – original draft; Nicolas De Neuter, Formal analysis, Investigation; Nina Keersmaekers, Data curation, Formal analysis; Arvid Suls, Formal analysis, Investigation, Methodology, Writing – review and editing; Hilde Jansens, Hans De Reu, Marie-Paule Emonds, Investigation, Methodology; Aisha Souquette, Data curation, Formal analysis, Investigation, Writing – review and editing; Evelien Smits, Geert Mortier, Supervision, Writing – review and editing; Eva Lion, Methodology, Writing – review and editing; Paul G Thomas, Funding acquisition, Methodology, Supervision, Writing – review and editing; Pierre Van Damme, Philippe Beutels, Conceptualization, Project administration, Supervision, Writing – review and editing; Kris Laukens, Conceptualization, Data curation, Funding acquisition, Project administration; Viggo Van Tendeloo, Conceptualization, Funding acquisition, Project administration, Writing – review and editing; Benson Ogunjimi, Conceptualization, Funding acquisition, Project administration, Supervision, Writing – review and editing

### Author ORCIDs

George Elias http://orcid.org/0000-0001-8419-9544
Pieter Meysman http://orcid.org/0000-0001-5903-633X
Nicolas De Neuter http://orcid.org/0000-0002-6011-6457
Benson Ogunjimi http://orcid.org/0000-0002-0831-2063

### Ethics

**Ethics**.This study was approved by the Ethics Committee of the Antwerp University Hospital and University of Antwerp, Belgium (IRB 15/19/210). Written informed consent and consent to publish were obtained from all study participants. All experiments and methods were performed in accordance with the relevant guidelines and regulations when applicable.

### Decision letter and Author response

Decision letter https://doi.org/10.7554/eLife.68388.sa1
Author response https://doi.org/10.7554/eLife.68388.sa2

## Additional files

### Supplementary files

• Supplementary file 1. List of 54 single peptides, each 15 amino acid (AA) long with an 11-AA overlap spanning the 226 AAs along the small S protein of hepatitis B (HB) surface antigen (HBsAg).

• Supplementary file 2. Overview of the single peptides tested for each vaccinee in the carboxyfluorescein succinimidyl ester (CFSE) assay.

• Transparent reporting form

### Data availability

The sequencing data that support the findings of this study have been deposited on Zenodo (https://doi.org/10.5281/zenodo.3989144). Flow Cytometry Standard (FCS) data files with associated FlowJo workspaces are deposited at flowrepository.org under the following experiment names: epitope mapping: https://flowrepository.org/id/FR-FCM-Z2TN; in vitro T cell expansion: https://flowrepository.org/id/FR-FCM-Z2TM; ex vivo CD4 T cell assay: https://flowrepository.org/id/FR-FCM-Z2TL.

The following dataset was generated:

| Author(s) | Year | Dataset title | Dataset URL | Database and Identifier |
|---|---|---|---|---|
| Meysman P | 2020 | Preexisting memory CD4 T cells in naïve individuals confer robust immunity upon vaccination | https://doi.org/10.5281/zenodo.3989144 | Zenodo, 10.5281/zenodo.3989144 |
| Elias G | 2020 | In vitro T cell expansion | https://flowrepository.org/id/FR-FCM-Z2TM | flowrepository, FR-FCM-Z2TM |
| Elias G | 2020 | ex vivo CD4 T cell assay | https://flowrepository.org/id/FR-FCM-Z2TL | flowrepository, FR-FCM-Z2TL |
| Elias G | 2020 | Epitope mapping | https://flowrepository.org/id/FR-FCM-Z2TN | flowrepository, FR-FCM-Z2TN |

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
