## [Editor Report]

By using modern high-throughput sequencing this paper demonstrates that the antibody mediated immune responses that are elicited by vaccination are improved by pre-existing memory CD4 T cell responses. The experimental data are an important contribution and would be useful as a data resource for future research. All reviewers agree that the findings are of great interest to the community.

---

## [Decision Letter]

**Decision letter after peer review:**

Thank you for submitting your article "Preexisting memory CD4 T cells in naïve individuals confer robust immunity upon hepatitis B vaccination" for consideration by *eLife*. Your article has been reviewed by 3 peer reviewers, and the evaluation has been overseen by a Reviewing Editor and Aleksandra Walczak as the Senior Editor. The following individuals involved in review of your submission have agreed to reveal their identity: Rob J de Boer (Reviewer #2); William S DeWitt III (Reviewer #3).

Essential revisions:

1) Improve the statistical analysis for comparing the early- and late-converters: The authors compare Shannon entropy for day 60 vs. 0 and claim that the difference is more significant in early converters. The p-value however is marginal and does not support such strong claim, especially after taking into account multiple test correction. Please improve this analysis.

2) Develop a randomization test and a null model to show that the expansion of Ag specific T cells from day 0 to 60 is indeed significant.

3) Discuss how the commonality or differences in HLA types may impact the conclusions of the paper.

4) Since you cross validation is done for the ROC curves, please add error bars to the plots as well.

5) Better clarify the procedure for R_hbs_ measure. What is the distance d? How is the distance cutoff parameter for epitope based classification is chosen? What are the statistics of bystander and non-bystander sequences? How robust are the results to this exact choice?

Ideally the authors should try more suitable TCR distance models like TCRdist by Dash et al. for this analysis.

6) Clarify the language throughout the manuscript as detailed by reviewer #2.

7) Better clarify the mathematical notations in the manuscript.

8) Improve the code in GitHub.

9) Improve the quality and resolution of the figures.

*Reviewer #1 (Recommendations for the authors):*

1. Please make images properly in pdf – one can't zoom in and sometimes axis labels are unreadable.

2. Since you are doing cross validation in the ROC curves, please add error bars to the plots as well.

3. Figure 5a: I guess you tried to look for significant difference for all possible cell subsets. Have you corrected the p-value for multiple testing?

4. Given that the a central result of the paper derives from the ratio R_hbs_ measure, more care must be given to properly explain what you are doing:

5. What is the distance d?

6. How do you set the free parameter c?

7. How many bystander and non-bystander sequences you have?

8. What do you get without the bystander normalisation (only numerator)?

9. Could you explain in more detail how do you select these bystander sequences?

10. Is this bystander normalisation already enough to classify early vs late converters or do you get also information from the numerator?

11. line162: can we understand this from the fact that late converters haven't yet built a proper response at day 60?

*Reviewer #2 (Recommendations for the authors):*

The manuscript is not well written and most of the readers will not easily understand what exactly has been done in the various analysis. Most of my review will therefore asking for clarification.

Generally the language is not very exact. A few examples from page 4:

"Antigen presentation via major histocompatibility complex (MHC) (encoded by HLA genes), together with the right costimulatory and cytokine signals, are responsible for T cell activation (Curtsinger and Mescher, 2010; Esensten et al., 2016)."

– I miss the word proteins after (MHC).

On line 90 you write "the highly degenerate nature of the CD4 T cell recognition" and above you wrote that specificity is imparted.

Line 142: we detected a significant increase in the TCR repertoire Shannon's entropy for early-converters (Figure 2a): this looks like a very minor difference in Figure 2a. Which test was performed and what is the effect size?

Line 143: please explain what you mean by "less clonal".

Line 153: measuring CFSE on day 60 and tracking clones from time point 0 to 60 probably means that you search for the clones that dilute CFSE on day 60 in the day 0 repertoires. This is not explained.

Line 156. "a significant increase in the frequency of unique HBsAg-specific TCR sequences": what is the frequency of a unique sequence? Its abundance? Why then the "unique"? Do you mean "a significant increase in the abundance of TCR sequences specific for HBsAg peptides"?

Line 167: "Thus, although we see a rise in the number of vaccine-specific TCR clonotypes from day 0 to day 60, this cannot be attributed to an expansion of preexisting TCR clonotypes but rather the recruitment of new TCR clonotypes" I don't see how the number of vaccine-specific TCR clonotypes could have increased by an expansion of preexisting TCR clonotypes.

Line 169: "rather the recruitment of new TCRB clonotypes (presumably from the naïve T cell compartment): how about memory clonotypes that were not present in the day 0 sample?

Line 200: Unclear sentence: "These classifications were integrated into a model which outputs a ratio Rhbs for any TCR repertoire representing the amount of HBsAg peptide-specific clonotypes". Is the ratio Rhbs predicting the fraction of HBsAg peptide-specific clonotypes in a repertoire? Note that ratio, amount and fraction would then have the same meaning.

Line 200-215 is poorly written, e.g.,

– This model applied to the memory repertoire at day 60 shows.

– To account for the age variable, a model in which.….

Line 234-237 Hard to read sentence.

*Reviewer #3 (Recommendations for the authors):*

There are some mathematical notation issues that make it difficult to understand the discriminative ratio R, as defined on line 591 in the methods. The function d(.,.) was previously defined as a hamming distance between sequences, but in this definition it takes arguments t_repi_ and t_pep_, each of which is defined as a set of sequences. My interpretation is that {t_repi_ | d(t_repi_, t_pep_) < c} means something more like {x ∈ t_repi_ | min_(y ∈ t_pep_) d(x, y) < c}.

The code on GitHub needs substantial improvements to documentation. I was unable to find the part of the code where the c parameter is set for R_hbs_, or where the epitope specific clones are held out for the individual being classified in the LOO procedure. I suggest expanding the readme to detail how to use each script, perhaps with example commands, and where various important methodological details are implemented.

Many figure panels are of such low resolution that axis labels and annotations are illegible.

I would expect that HLA type would strongly influence the inter-individual relevance of the peptide specific TCRs, and the performance of this classifier (especially between individuals with different genetic backgrounds). Can the authors comment on why this wasn't an issue in this study?

[Editors' note: further revisions were suggested prior to acceptance, as described below.]

Thank you for resubmitting your article "Preexisting memory CD4 T cells in naïve individuals confer robust immunity upon hepatitis B vaccination" for consideration by *eLife*. Your article has been reviewed by 3 peer reviewers, and the evaluation has been overseen by a Reviewing Editor and Aleksandra Walczak as the Senior Editor. The following individuals involved in review of your submission have agreed to reveal their identity: Giulio Isacchini (Reviewer #1); Rob J de Boer (Reviewer #2); William S DeWitt III (Reviewer #3).

By using modern high-throughput sequencing this paper demonstrates the antibody mediated immune responses that are elicited by vaccination are improved by pre-existing memory CD4 T cell responses. The experimental data are an important contribution and would be useful as a data resource for future research. All reviewers agree that the findings are great interest and the revision has addressed all the previous concerns.

Essential revisions:

1) Figure 1—figure supplement 1: early converters subfigure. Comparison between 180 and 365 does not look significant. Is the **** indication in the figure a mistake?

*Reviewer #1 (Recommendations for the authors):*

The revised manuscript has considerably improved in quality and is ready for publication. The authors have clearly responded to the observations raised by the reviewers. The paper describes an original and important study and is of interest for *eLife* readers.

*Reviewer #2 (Recommendations for the authors):*

I have read the rebuttal letter and find that the authors have responded well to my suggestions.

*Reviewer #3 (Recommendations for the authors):*

The authors' responses to my comments on their initial submission are very thorough, and all of my concerns have been adequately addressed. They have improved the rigor of several statistical analyses, and clarified presentation of technical aspects that had previously been vague or confusing. They have also improved the open source code repository, adding more clear documentation of where key methods from the paper are implemented.

The findings of the manuscript are convincing, and the data is a valuable resource. I have no further concerns.

---

## [Author Response]

Essential revisions:1) Improve the statistical analysis for comparing the early- and late-converters: The authors compare Shannon entropy for day 60 vs. 0 and claim that the difference is more significant in early converters. The p-value however is marginal and does not support such strong claim, especially after taking into account multiple test correction. Please improve this analysis.

This analysis has been removed and replaced with a more robust analysis based on the null model that has been added (see point #2).

2) Develop a randomization test and a null model to show that the expansion of Ag specific T cells from day 0 to 60 is indeed significant.

A null model has been added based on additional experimental data derived from VZV-specific T cells, which show no increase between day 0 and day 60.

3) Discuss how the commonality or differences in HLA types may impact the conclusions of the paper.

A section on the impact of HLA types has been added to the discussion.

4) Since you cross validation is done for the ROC curves, please add error bars to the plots as well.

Error bars have been added to the ROC curve where they remain visually distinct.

5) Better clarify the procedure for R_hbs_ measure. What is the distance d? How is the distance cutoff parameter for epitope based classification is chosen? What are the statistics of bystander and non-bystander sequences? How robust are the results to this exact choice?

The explanation with regards to the Rhbs metric has been reworked in the revised manuscript based on the comments and suggestions.

Ideally the authors should try more suitable TCR distance models like TCRdist by Dash et al. for this analysis.

We have clarified our choice of distance model and referred to relevant literature.

6) Clarify the language throughout the manuscript as detailed by reviewer #2.

The reviewer suggestions with regards to language usage have been addressed. The manuscript has been carefully reread and any additional problems are hopefully now resolved.

7) Better clarify the mathematical notations in the manuscript.

The mathematical notation has been updated based on the reviewer comment.

8) Improve the code in GitHub.

The code base on GitHub has been substantially reworked. It now features a step-by-step help file and the script have been documented to refer to the manuscript definitions where possible.

9) Improve the quality and resolution of the figures.

The low quality of the figures is due to the conversion of the word document to PDF format. We will upload the source figure files separately in the submission of the revised manuscript.

Reviewer #1 (Recommendations for the authors):1. Please make images properly in pdf one can't zoom in and sometimes axis labels are unreadable.

We apologize for this inconvenience. The low quality of the figures is due to the conversion of the word document to PDF format by the manuscript handling system. We will upload the source figure files separately in the submission of the revised manuscript in the hopes that this will result in higher quality images.

2. Since you are doing cross validation in the ROC curves, please add error bars to the plots as well.

We have added a confidence interval to the ROC curves. However, we refrained from doing so for 4e and 4f as they feature three ROC curves simultaneously and comprehension becomes difficult with the confidence intervals.

3. Figure 5a: I guess you tried to look for significant difference for all possible cell subsets. Have you corrected the p-value for multiple testing?

Multiple testing correction is implemented throughout the manuscript. This has now been clarified in the manuscript text.

4. Given that the a central result of the paper derives from the ratio R_hbs_ measure, more care must be given to properly explain what you are doing:

Several sections detailing the ratio R_hbs_ measure have been clarified.

5. What is the distance d?

The distance d is defined as the Hamming distance between TCR CDR3 amino acid sequences. This has been clarified in the revised manuscript.

6. How do you set the free parameter c?

We derive c from Meysman et al. (2018, Bioinformatics), where a value of 1 was found to be the most optimal threshold on a validation data set. This has been clarified in the revised manuscript.

7. How many bystander and non-bystander sequences you have?

This is visualized in figure 3B which list the numerator and denominator values for each sample in the data set. This fact has been clarified in the caption.

8. What do you get without the bystander normalisation (only numerator)?

This can be inferred from figure 3B. The numerator alone provides a clear separation but is clearly inflated by the false positives. This has been clarified in the results and figure legend.

9. Could you explain in more detail how do you select these bystander sequences?

The relevant methods and Results section has been extended.

10. Is this bystander normalisation already enough to classify early vs late converters or do you get also information from the numerator?

See previous answer, the numerator provides most of the information with respect to the early converters. This has been clarified in the result text.

11. line162: can we understand this from the fact that late converters haven't yet built a proper response at day 60?

Indeed, at this time point, the late converters have not established a proper measurable T-cell response/antibody titer.

Reviewer #2 (Recommendations for the authors):The manuscript is not well written and most of the readers will not easily understand what exactly has been done in the various analysis. Most of my review will therefore asking for clarification.Generally the language is not very exact.

Thank you for bringing this to our attention. We have endeavored to clarify the language throughout the manuscript.

A few examples from page 4:"Antigen presentation via major histocompatibility complex (MHC) (encoded by HLA genes), together with the right costimulatory and cytokine signals, are responsible for T cell activation (Curtsinger and Mescher, 2010; Esensten et al., 2016)."– I miss the word proteins after (MHC).

The word “protein” has been added.

On line 90 you write "the highly degenerate nature of the CD4 T cell recognition" and above you wrote that specificity is imparted.

These two sentences have now been corrected. They now read:

“The T-cell receptor (TCR) αβ heterodimer binds the peptide-MHC (pMHC) complex and confers the specificity of a T-cell to an epitope.”

“… as the encounter with environmentally-derived peptides activates cross-reactive T cells due to the flexible nature of the CD4 T-cell recognition of peptide-MHC complex.”

Line 142: we detected a significant increase in the TCR repertoire Shannon's entropy for early-converters (Figure 2a): this looks like a very minor difference in Figure 2a. Which test was performed and what is the effect size?

This section has been adapted with regards to the Shannon equitability index. In short, the prior findings were not robust to this change in definition. This is in line with what the reviewer indicates were a ‘very minor difference’. As they had no impact on the conclusions of the paper, they have been removed.

Line 143: please explain what you mean by "less clonal".

This has been clarified to mean that a set of clonotypes have likely expanded in relative frequency.

Line 153: measuring CFSE on day 60 and tracking clones from time point 0 to 60 probably means that you search for the clones that dilute CFSE on day 60 in the day 0 repertoires. This is not explained.

We agree with the reviewer that this was unclear in the original manuscript. This point has been clearly explained in the revised manuscript.

Line 156. "a significant increase in the frequency of unique HBsAg-specific TCR sequences": what is the frequency of a unique sequence? Its abundance? Why then the "unique"? Do you mean "a significant increase in the abundance of TCR sequences specific for HBsAg peptides"?

The term “increase in frequency of unique sequences” was an amalgam of terms, and has been replaced with the “increase in breadth”.

Line 167: "Thus, although we see a rise in the number of vaccine-specific TCR clonotypes from day 0 to day 60, this cannot be attributed to an expansion of preexisting TCR clonotypes but rather the recruitment of new TCR clonotypes" I don't see how the number of vaccine-specific TCR clonotypes could have increased by an expansion of preexisting TCR clonotypes.

This was indeed incorrectly worded and has been corrected.

Line 169: "rather the recruitment of new TCRB clonotypes (presumably from the naïve T cell compartment): how about memory clonotypes that were not present in the day 0 sample?

This sentence has been rewritten to clarify that we indeed mean the contrast between the memory repertoires from day 0 and day 60. The postulated origin of these clonotypes has been removed to avoid unnecessary assumptions in the Results section.

Line 200: Unclear sentence: "These classifications were integrated into a model which outputs a ratio Rhbs for any TCR repertoire representing the amount of HBsAg peptide-specific clonotypes". Is the ratio Rhbs predicting the fraction of HBsAg peptide-specific clonotypes in a repertoire? Note that ratio, amount and fraction would then have the same meaning.

This section now details a stricter definition to avoid confusion. We have also opted to use the term ‘breadth’ were appropriate to be specific which parameters are being used.

Line 200-215 is poorly written, e.g.,– This model applied to the memory repertoire at day 60 shows.– To account for the age variable, a model in which.….

This section has been substantially rewritten.

Line 234-237 Hard to read sentence.

The sentence has not been modified to make it clearer. CD25+CD127^−^ and CD25^−^CD27+ phenotype can be used to identify T_REG_ and T_CON_ cells and in this case, we show that CD40L^+^4-1BB^−^ and CD40L^−^4-1BB^+^ CD4 T as captured by activation markers do agree with this classification and hence support the use of the converse expression of CD40L and 4-1BB as signature of the two CD4 T cell phenotypes.

Reviewer #3 (Recommendations for the authors):There are some mathematical notation issues that make it difficult to understand the discriminative ratio R, as defined on line 591 in the methods. The function d(.,.) was previously defined as a hamming distance between sequences, but in this definition it takes arguments t_repi_ and t_pep_, each of which is defined as a set of sequences. My interpretation is that {t_repi_ | d(t_repi_, t_pep_) < c} means something more like {x ∈ t_repi_ | min_(y ∈ t_pep_) d(x, y) < c}.

The reviewer is correct is their interpretation of the discriminative ratio R. The suggested alterations to the formula have been implemented in the text.

The code on GitHub needs substantial improvements to documentation. I was unable to find the part of the code where the c parameter is set for R_hbs_, or where the epitope specific clones are held out for the individual being classified in the LOO procedure. I suggest expanding the readme to detail how to use each script, perhaps with example commands, and where various important methodological details are implemented.

We agree with the reviewer that the previous code base lacked proper documentation. We have now greatly extended this with a step-by-step guide and additional comments. In addition, we have rewritten the code to be more legible, while confirming that no results have been impacted. In particular, the R_hbs_ parameters are now explicitly marked in the code where they are generated. At the reviewers request, the c parameter has been extracted from the inner workings of the hamming distance calculator to a parameter that is set at the start of the code (within peptide.py).

Many figure panels are of such low resolution that axis labels and annotations are illegible.

We apologize for this inconvenience. The low quality of the figures is due to the conversion of the word document to PDF format by the manuscript handling system. We will upload the source figure files separately in the submission of the revised manuscript in the hopes that this will result in higher quality images.

I would expect that HLA type would strongly influence the inter-individual relevance of the peptide specific TCRs, and the performance of this classifier (especially between individuals with different genetic backgrounds). Can the authors comment on why this wasn't an issue in this study?

This is an important point raised by the reviewer. We have added a comment on this fact to the discussion, it reads:

“This is a non-trivial task as the MHC class II molecules vary between vaccinees, which in turn allows for great variation of the presented epitopes. However, we were able to cover the entire HBsAg and obtained a sufficient sample size to likely cover the most immunoprevalent epitopes. It can be presumed that we do not capture the full HBsAg T cell response, but obtain enough sequences which are representative for the response as a whole.”

[Editors' note: further revisions were suggested prior to acceptance, as described below.]

Essential revisions:1) Figure 1—figure supplement 1: early converters subfigure. Comparison between 180 and 365 does not look significant. Is the **** indication in the figure a mistake?

We revisited the script. The comparison of the antibody titer between day 180 and 365 for early-converters is significantly different and it is not a typo. This is a paired analysis and indeed, as shown in Author response image 1, the decrease in antibody titer is consistent for all vaccinees between day 180 and day 365 while the increase between day 60 and day 180 is not.

**Author response image 1. sa2fig1:** 

**Author response table 1. sa2table1:** 

Median of antibody titer over time				
Status	Time_Point	Antibody_titre_median		
Early-converter	0	2		
Early-converter	60	82.23		
Early-converter	180	167.7		
Early-converter	365	62.01		
				
Statistical analysis				
group1	group2	p	p.adj	p.adj.signif
0	60	9.54E-07	4.77E-06	****
60	180	0.785	1	ns
180	365	9.54E-07	4.77E-06	****
0	180	9.54E-07	4.77E-06	****
0	365	9.54E-07	4.77E-06	****